# A cellular trafficking signal in the SIV envelope protein cytoplasmic domain is strongly selected for in pathogenic infection

Scott P. Lawrence[1‡], Samra E. Elser[2‡], Workineh Torben[3¤a‡], Robert V. Blair[3], Bapi Pahar[3¤b], Pyone P. Aye[3], Faith Schiro[3], Dawn Szeltner[3], Lara A. Doyle-Meyers[3], Beth S. Haggarty[2], Andrea P. O. Jordan[2], Josephine Romano[2], George J. Leslie[2], Xavier Alvarez[3¤c], David H. O'Connor[4], Roger W. Wiseman[4], Christine M. Fennessey[5], Yuan Li[5], Michael Piatak, Jr[5†], Jeffrey D. Lifson[5], Celia C. LaBranche[6], Andrew A. Lackner[3†], Brandon F. Keele[5], Nicholas J. Maness[3], Mark Marsh[1]*, James A. Hoxie[2]*

**1** MRC Laboratory for Molecular Cell Biology, University College London, London, United Kingdom, **2** Perelman School of Medicine, University of Pennsylvania, Philadelphia, Pennsylvania, United States of America, **3** Tulane National Primate Research Center, Covington, Louisiana, United States of America, **4** Wisconsin National Primate Research Center, Madison, Wisconsin, United States of America, **5** AIDS and Cancer Virus Program, Frederick National Laboratory for Cancer Research, Frederick, Maryland, United States of America, **6** Duke University Medical Center, Durham, North Carolina, United States of America

† Deceased.
¤a Current address: Louisiana State University at Alexandria, Alexandria, Louisiana, United States of America
¤b Current address: Trudeau Institute, Saranac Lake, New York, United States of America
¤c Current address: Southwest National Primate Research Center, San Antonio, Texas, United States of America
‡ These authors share first authorship on this work.
* m.marsh@ucl.ac.uk (MM); hoxie@pennmedicine.upenn.edu (JAH)

**Data Availability Statement:** All sequences have been submitted to Genebank; Accession numbers BankIt2566664: ON096336 - ON097106.

## Abstract

The HIV/SIV envelope glycoprotein (Env) cytoplasmic domain contains a highly conserved Tyr-based trafficking signal that mediates both clathrin-dependent endocytosis and polarized sorting. Despite extensive analysis, the role of these functions in viral infection and pathogenesis is unclear. An SIV molecular clone (SIVmac239) in which this signal is inactivated by deletion of Gly-720 and Tyr-721 (SIVmac239ΔGY), replicates acutely to high levels in pigtail macaques (PTM) but is rapidly controlled. However, we previously reported that rhesus macaques and PTM can progress to AIDS following SIVmac239ΔGY infection in association with novel amino acid changes in the Env cytoplasmic domain. These included an R722G flanking the ΔGY deletion and a nine nucleotide deletion encoding amino acids 734–736 (ΔQTH) that overlaps the *rev* and *tat* open reading frames. We show that molecular clones containing these mutations reconstitute signals for both endocytosis and polarized sorting. In one PTM, a novel genotype was selected that generated a new signal for polarized sorting but not endocytosis. This genotype, together with the ΔGY mutation, was conserved in association with high viral loads for several months when introduced into naïve PTMs. For the first time, our findings reveal strong selection pressure for Env endocytosis and particularly for polarized sorting during pathogenic SIV infection *in vivo*.

**Funding:** The project has been supported by the following funding: Federal funds from 1) the Office of Research Infrastructure Programs, National Institutes of Health, TNPRC base grant P51OD011104; 2) the National Cancer Institute, National Institutes of Health, under Contract Nos. HHSN261200800001E and 75N91019D00024; 3) the National Institute of Allergy and Infectious Diseases, National Institutes of Health, P30 AI45008 to the Penn Center for AIDS Research and R01 AI138782 to JAH. MM and SPL were supported by UK Medical Research Council funding to the MRC-UCL Laboratory for Molecular Cell Biology University Unit (MC_UU00012/1 and MC_U12266B). Pigtail macaques acquired from Johns Hopkins University were supported by NIH grant 5U42OD013117 prior to sale to Tulane. The funders had no role in study design, data collection and analysis, decision to publish, or preparation of the manuscript.

**Competing interests:** The authors have declared that no competing interests exist. Authors Michael Piatak Jr. and Andrew A Lackner are deceased. On their behalf, the corresponding authors have reported their contributions to the best of their knowledge.

## Author summary

In the cytoplasmic domain (CD) of their envelope glycoproteins (Env), all human and simian immunodeficiency viruses (HIV and SIV) have a tyrosine-dependent motif that is a potent signal for Env endocytosis and polarized sorting in infected cells. The role and relevance of this trafficking signal in pathogenesis is unknown. For a pathogenic SIV, we have shown that deletion of this tyrosine and a preceding glycine, creating a virus termed ΔGY, results in profoundly altered pathogenesis in pigtail macaques. Viral replication is rapidly suppressed, systemic immune activation fails to occur, and CD4 cells are spared. However, although uncommon, ΔGY-infected macaques can develop high viral loads and disease in association with novel mutations in the Env CD. Here we show that these mutations restore trafficking functions, remarkably, at the expense of *tat* and *rev* genes in overlapping reading frames. We quantified the effects of these new signals and demonstrated their ability to confer high viral loads and/or disease to ΔGY-infected animals. These findings demonstrate strong *in vivo* selection pressures to maintain Env trafficking, in particular, polarized sorting. We present hypotheses for why this long recognized, but poorly understood Env function is conserved and essential for HIV/SIV pathogenesis.

## Introduction

Central themes that underlie pathogenesis of human immunodeficiency virus type-1 (HIV) in humans, and simian immunodeficiency virus (SIV) in Asian macaques, include host failure to control viral replication, chronic immune activation, and progressive loss of CD4+ T cells [1–3]. Although AIDS can take years to develop, there are critical early events that dictate the outcome of infection. CD4+/CCR5+ T cells in gut-associated lymphoid tissue (GALT) are rapidly and massively depleted leading to compromised epithelial barrier function, microbial translocation, and systemic immune activation [1–5]. In addition, within hours of SIV infection, pathological innate immune sensors are engaged that dysregulate antiviral interferon responses and initiate proinflammatory programs that are sustained and may compromise subsequent adaptive immune responses [6–10]. Nevertheless, immune control of viral replication can occur, but the mechanisms and determinants for this outcome are poorly understood [11–13].

HIV/SIV Env is expressed on infected cells and on virions as trimers of gp120/gp41 heterodimers in which gp41 anchors Env to cellular and viral membranes. Cellular infection is initiated when viral gp120 binds to CD4 and a coreceptor (CCR5 or CXCR4), leading to gp41-mediated membrane fusion [14]. A conserved feature of gp41 is a long cytoplasmic domain (CD; ~160 amino acids [a.a.]) containing motifs that engage cellular trafficking and signalling pathways [15,16]. We have shown that deletion of Gly and Tyr (ΔGY; a.a. 720 and 721) from a highly conserved, membrane-proximal, GYxxØ-type motif (x = any a.a.; Ø = a.a. with a bulky hydrophobic side chain) [17] in the SIVmac239 Env CD (GYRPV), creates a virus (termed SIVmac239ΔGY) that in PTM leads to acute infection, with viral RNA peaks similar to parental SIVmac239, followed by control (<15–50 RNA copies/ml) with onset of host cellular immune responses [18]. In contrast to SIVmac239, in SIVmac239ΔGY infection CD4+ T cells are not depleted in blood or gut, GALT infection is only transient, there is no microbial translocation or chronic immune activation, and animals remain healthy for months to years as elite controllers [18]. Control of SIVmac239ΔGY is independent of neutralizing antibodies but associated with strong, polyfunctional antiviral CD4+ and CD8+ T cell responses with a role for CD8+ CTL or NK cells shown by anti-CD8 cell depletion [18]. Interestingly, in rhesus

macaques (RM), control of SIVmac239ΔGY infection is incomplete, and animals progress to disease with detectable viral loads. The fact that PTM can suppress viral replication completely is paradoxical, given that SIVmac239 infected PTM typically progress to AIDS more rapidly than RM [19].

For HIV/SIV Env, the GYxxØ motif, together with more variable C-terminal di-leucine motifs, influences the expression and distribution of Env on infected cells by engaging clathrin-based endocytosis [17,20–22] and contributes to the well-recognized paucity of Env on virions [22–24]. The GYxxØ motif has also been shown to direct polarized sorting of Env to the basal and lateral plasma membranes in MDCK and Vero cells [25–28]. Polarized sorting of HIV Env has been proposed to influence the distribution of viral Gag protein in T cells during viral budding (25), though Gag can also be sorted independently of Env [29–31]. Although the GYxxØ motif is highly conserved (15), the requirement and roles for Env endocytosis and polarized sorting *in vivo* are unclear, as is the mechanism(s) through which the ΔGY deletion has such a profound impact on pathogenesis.

Here we evaluated the effects of the ΔGY deletion on viral assembly and replication *in vitro* and assessed the effects of previously reported mutations acquired in SIVmac239ΔGY-infected RM and PTM that progressed to disease [32,33]. These mutations encode an R722G substitution flanking the ΔGY deletion and loss of 3 downstream amino acids (ΔQTH; a.a. 734–736). We show that R722G restored a ΔGY-associated decrease in Env content in infected cells and on virions. In contrast, ΔQTH generated new Tyr-dependent signals for both endocytosis and polarized sorting. When introduced into SIVmac239ΔGY, these changes partially restored pathogenesis in PTM. In an additional animal inoculated with SIVmac239ΔGY containing R722G that progressed to AIDS, 3 substitutions appeared in the Env CD that generated a new signal for polarized sorting but not endocytosis. When introduced into SIVmac239ΔGY with R722G and inoculated into PTM, all 3 substitutions were retained and associated with sustained intermediate to high levels of viremia.

Our results indicate that reduction of Env on virions and loss of cellular trafficking functions caused by the ΔGY deletion could be restored *in vivo* by acquisition of novel compensatory mutations and that these functions correlated with a gain of pathogenicity. Our findings reveal, for the first time, strong selection pressures to maintain polarized trafficking of Env *in vivo* and demonstrate that loss of this function can lead to potent host immune control reminiscent of elite HIV control in humans.

## Results

### Mutations acquired during pathogenic ΔGY infection

We previously reported that 4 of 4 RM [32] and 2 of 21 PTM [18] infected with SIVmac239ΔGY progressed to AIDS. Single genome amplification and sequencing (SGS) revealed that the ΔGY deletion was maintained in all viral amplicons in plasma from these animals and identified novel mutations in the Env CD. These included either an R722G substitution flanking the ΔGY deletion, which restored a Gly at position 720, or an S727P substitution (Fig 1). In 2 RM, acquisition of R722G was followed by deletions of 9 nucleotides (nt 8803–8811 in animal DT18 and nt 8804–8812 in animal DD84), each of which removed a.a. 734–736 (QTH) generating YFQI and YFQL, respectively. These latter mutations are unique among SIVmac-related sequences in the Los Alamos HIV Sequence Database and remarkable in that (i) they generated a YxxØ sequence reminiscent of the conserved GYRPV motif disrupted by the ΔGY deletion, and (ii) occurred within overlapping reading frames for the second exons of *tat* and *rev* (S1 Fig). S727P had been seen in an earlier study of SIVmac239ΔGY infection in RM [34] and was shown to increase infection in gut CD4+ T cells during acute infection [33]. Given

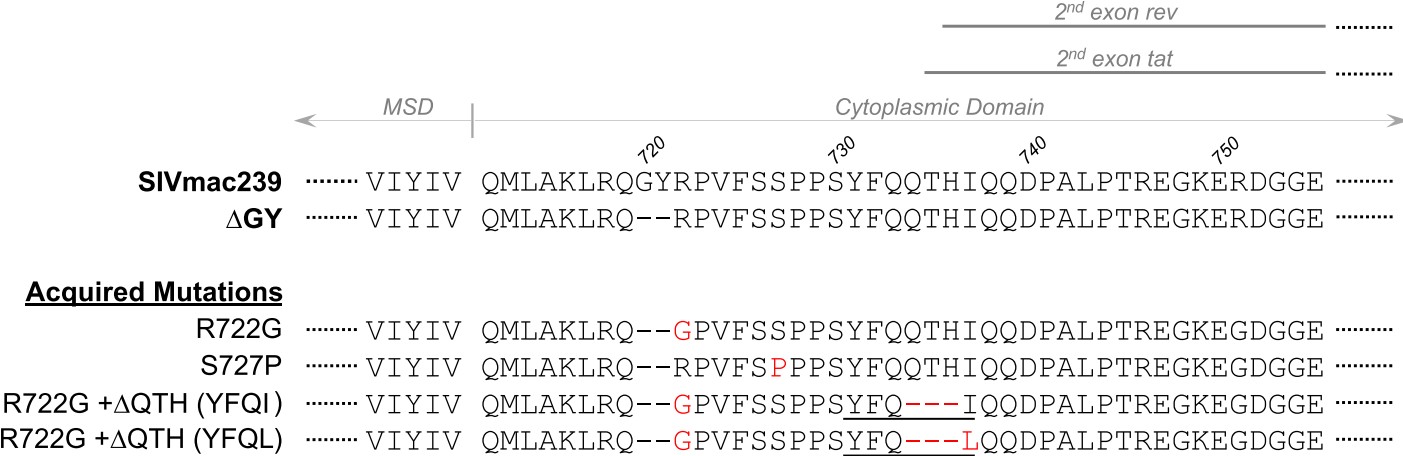

**Fig 1. Amino acid changes acquired in Env during pathogenic SIVmac239ΔGY infection.** Amino acid (a.a.) sequences of SIVmac239 and ΔGY Env are shown indicating the membrane spanning domain (MSD), the predicted start of the cytoplasmic domain, and approximate start sites for the second exons of *tat* and *rev* in overlapping reading frames. Below (in red) are a.a. changes previously reported in RM [32] and PTM [18] that failed to control ΔGY and progressed to disease. YxxØ motifs created by ΔQTH mutations are underlined (see S1 Fig for nt sequences).

that R722G, ΔQTH or S727P were seen in all 6 SIVmac239ΔGY-infected animals that progressed to disease [18,32], we evaluated the impact of these changes on Env expression and trafficking *in vitro* and on pathogenesis *in vivo*.

**Effects of ΔGY and changes acquired *in vivo* on Env expression.** To characterize the effects of the ΔGY deletion and the *in vivo* acquired a.a. changes, we first assessed Env expression on infected cells and virions (Fig 2). To avoid any variation in particle infectivity due to different SIV Envs, we used VSV-G pseudotyped SIVs. Cell lines lacking CD4 were used to avoid cytopathic effects associated with Env expression.

**Effects of the ΔGY deletion on Env content in cells and on virions.** Relative to p57/p27 Gag, ΔGY Env content in total cell lysates and on the surface of rhesus LLC-MK2 cells was reduced ~40% compared to SIVmac239 Env (Fig 2A and 2B). The decrease in Env was similar for both mature Env (gp120/gp41) and gp160, indicating that Env cleavage was unaltered. Relative to p27 Gag, ΔGY Env on virions was also reduced ~40% compared to SIVmac239 Env (Fig 2C). Similar results were seen for gp120 and gp41, indicating that Env shedding could not explain this difference. Gp160 on SIVmac239 virions was negligible but was increased slightly on ΔGY virions (from 3% to 11% of the total Env [p = 0.0193]). Thus, the reduced level of ΔGY Env in cells resulted in a corresponding decrease of ΔGY Env in virions.

**Effects of ΔGY-associated mutations acquired *in vivo* on Env content in cells and virions.** The a.a. changes shown in Fig 1 were introduced into SIVmac239ΔGY and Env levels in cells and virions determined as above. Strikingly, the reduction in Env content (gp120 and gp160) caused by the ΔGY deletion was rescued by R722G to levels equal to or exceeding total cell-associated and cell surface SIVmac239 Env (Fig 2A and 2B). R722G also increased ΔGY Env on virions and in cells when it contained the ΔQTH deletion (producing YFQL). On virions, this increase was predominantly due to gp160, suggesting less efficient cleavage of Env prior to incorporation into virions. Indeed, introduction of R722G into ΔGY Env resulted in a small reduction of Env processing (18.7% ± 4.5, p = 0.02) when compared to SIVmac239-infected cells. S727P also restored ΔGY Env content in infected cells to SIVmac239 levels (Fig 2A–C), but had a negligible effect on virion Env levels. In contrast, a ΔQTH deletion (producing YFQL) reduced total cell-associated, cell surface and virion Env when introduced into ΔGY (Fig 2A–C). However, the combination of R722G+ΔQTH (producing YFQL) generated

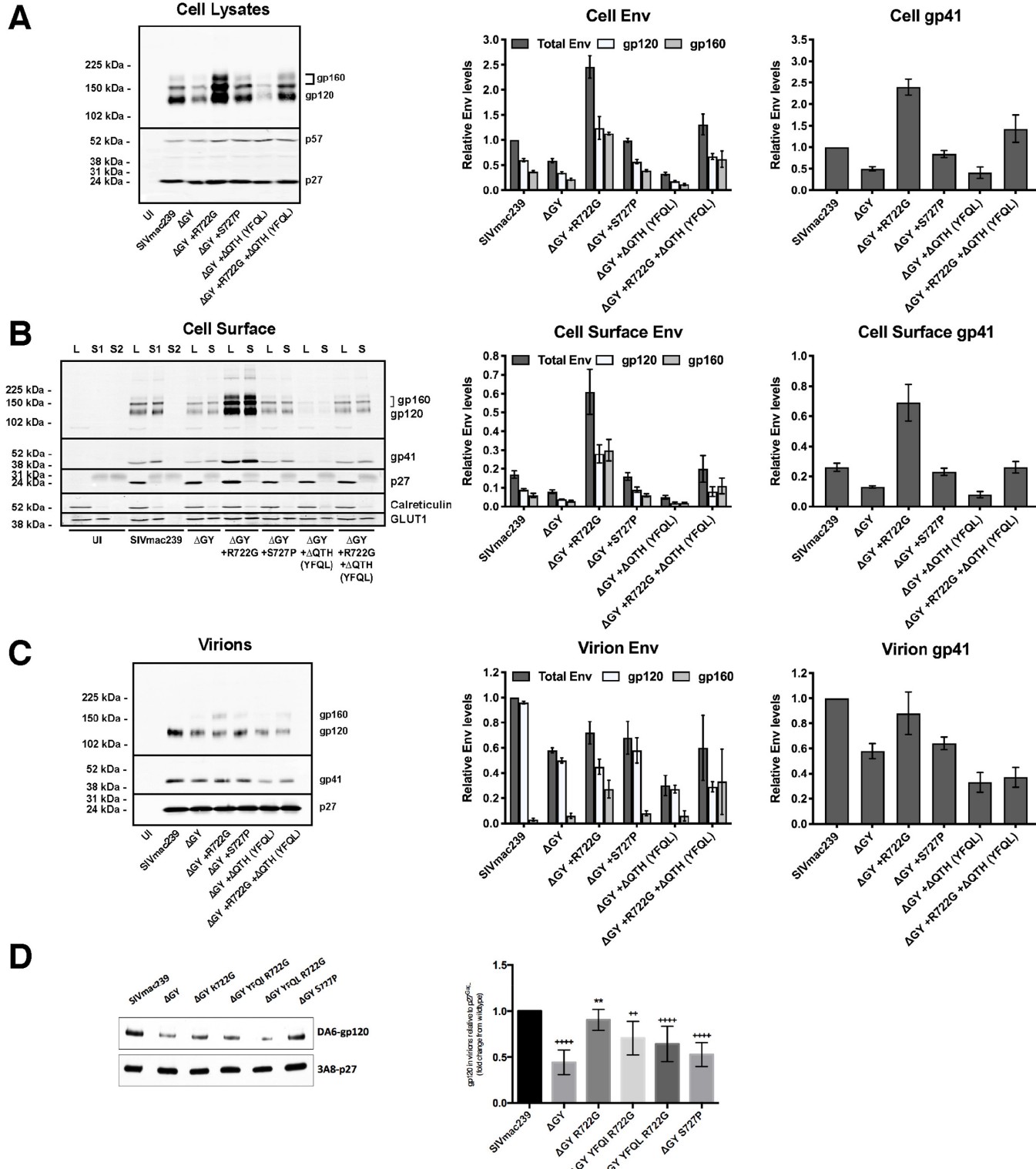

**Fig 2. The ΔGY deletion and changes acquired *in vivo* modulate Env content on cells and virions.** Expression of total cell, cell surface and virion-associated Env was determined by western blotting using LLC-MK2 cells infected with viruses encoding the indicated Env. **(A)** Total cell associated Env in cell lysates. **(B)** Cell surface Env isolated by biotinylation and streptavidin pull-down (S = Surface); cell lysates (L = lysate) correspond to 20% of the pull-down input. A second pull-down (S2) was performed to show that all biotinylated proteins were captured in the first round. **(C)** Env associated with virions released into supernatant

from cells shown in Panel A. **(D)** Env associated with virions produced from infected rhesus macaque PBMCs. Left panels show representative western blots; right panels show quantitation. Env levels are normalized to cell p57/p27 and relative to SIVmac239 (set at 1). The intensity of the processed (gp120) or unprocessed Env (gp160) is shown as the fraction of the total Env signal for each lane and relative to SIVmac239. Graphs display the mean ± SEM from n ≥ 3 independent experiments.

cell-associated, cell surface and virion Env levels similar to those seen with SIVmac239. Comparable results were seen in HEK293T cells and the human T lymphoid cell lines BC7 and CEMx174.

## Effects of ΔGY-associated mutations on Env content on virions produced in primary T cells

To determine the effects of ΔGY and acquired Env changes on virions produced in primary macaque lymphocytes, activated rhesus PBMCs were infected with viruses containing SIV-mac239 or ΔGY Envs, or ΔGY Envs containing R722G, R722G+ΔQTH (producing either YFQI or YFQL), or S727P. Virions were harvested after 4 days and analyzed for Env content by western blotting. As shown in Fig 2D, relative to virion p27 Gag, ΔGY Env was reduced ~50% compared to SIVmac239, similar to the reduction seen in virus produced from LLC-MK2 cells (Fig 2B). As in LLC-MK2 cells, R722G restored levels of Env containing the ΔGY deletion to near SIVmac239 levels, while S727P had a lesser effect.

Thus, while ΔGY decreased Env in cells, on the surface of infected cells and on virions, this defect was largely corrected by R722G. Although ΔQTH reduced Env on cells and virions, when combined with R722G, cellular and viral Env were restored to levels similar to SIVmac239.

## Alterations in the cellular distribution of Env caused by the ΔGY deletion and acquired mutations

ΔGY ablates a trafficking signal for clathrin-dependent endocytosis with the potential to alter the cellular distribution of Env [17,20–22]. To assess the effects of ΔGY and the mutations acquired *in vivo*, we used previously described chimeric reporters that contain the CD4 ecto- and membrane spanning domains (MSD) fused to the SIV Env CD (Fig 3A) [20,22]. SIV and HIV Envs contain additional endocytosis signals distal to the GYxxØ motif [20,22,35]; to avoid this additional trafficking information, these constructs contained only the membrane proximal 30 a.a. of the SIVmac Env CD. Stable HeLa cell lines were generated expressing the CD4-based reporters containing SIVmac239 or ΔGY CDs with or without the changes described in Fig 1.

The steady state distribution of each construct was evaluated on fixed and permeabilized cells by immunofluorescence microscopy. The chimera containing the SIVmac239 CD had an intracellular perinuclear distribution, while the construct containing the ΔGY CD was diffusely distributed on the cell surface (Figs 3B and S2C). A ΔGY CD containing ΔQTH, restored the perinuclear pattern, while chimeras containing only R722G or S727P remained diffusely distributed on the cell surface. To determine if any of these constructs trafficked to the cell surface and were then internalized, cells were incubated with anti-CD4 antibody prior to fixation and permeabilization. Cells expressing the SIVmac239 CD chimera exhibited prominent punctate intracellular staining showing that the protein had been exposed on the cell surface and then endocytosed, while the ΔGY CD chimera remained predominantly on the cell surface (Figs 3C and S2C). However, when this later construct contained the ΔQTH deletions, perinuclear punctate labelling was again seen, while the addition of R722G or S727P substitutions

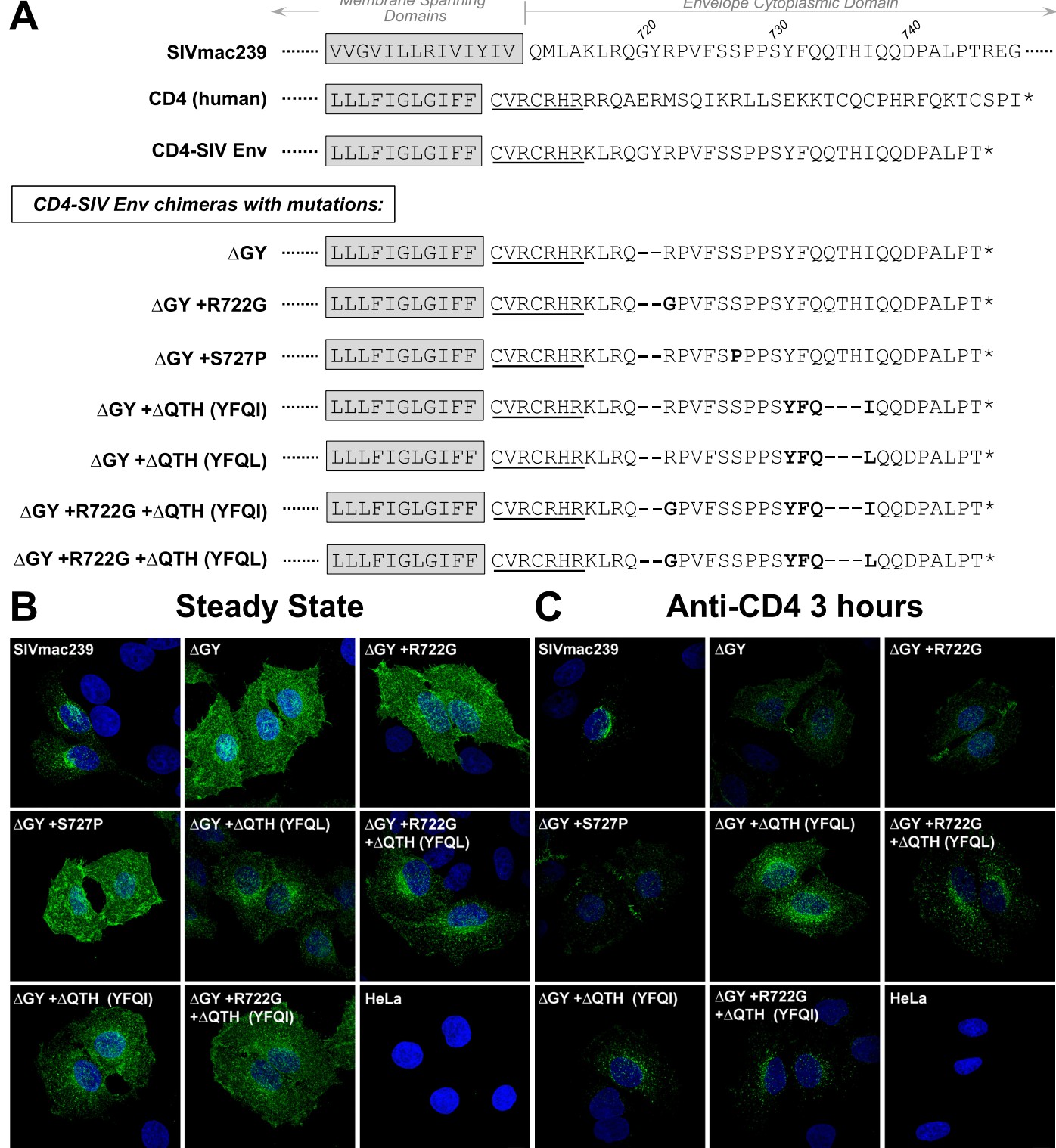

**Fig 3. Alterations in cellular trafficking induced by ΔGY and *in vivo* acquired mutations. (A)** Partial sequences for SIVmac239 Env and human CD4 are shown with the membrane spanning domain (MSD, shaded) and CD (partial for SIVmac239 Env and full length for CD4). A CD4-SIV Env CD construct is shown containing the CD4 ecto- and MSD with 7 a.a. from the membrane proximal cytoplasmic domain of CD4 (underlined) fused to a.a. 716–745 from the SIVmac239 Env CD. CD4-SIV Env CD chimeras are shown containing ΔGY and indicated changes (**bold**). **(B)** Steady state distribution of the CD4-SIV Env CD chimeras in stably transformed HeLa cells analysed by immunofluorescence microscopy using an anti-CD4 antibody. **(C)** Cellular distribution of CD4-SIV Env CD chimeras following incubation of cells with anti-CD4 antibody (Q4120; 3 hr at 37˚C) prior to fixation. Confocal Z stacks were deconvolved and displayed as maximum projections. Size bars = 10 μm.

had no effect. Collectively, these findings indicate that the endocytic trafficking function of the SIV CD ablated by ΔGY was restored by ΔQTH, but not by R722G or S727P.

### ΔQTH deletions create novel Tyr-based endocytosis signals

We quantified the effects of the ΔQTH deletions on the endocytic properties of the CD4-SIV Env CD constructs shown in Fig 3 by measuring the rate of uptake of an anti-CD4 antibody using a modification of a previously described protocol [22]. Cells were incubated with antibody at 4°C, washed and warmed to 37°C, and the decrease in cell-surface antibody measured over time (Fig 4). Endocytosis of the SIVmac239 CD construct followed a biphasic pattern, as previously described [22]; during an early rapid phase (0–5 min), when recycling is negligible, the SIVmac239 Env CD construct was internalized at ~12% per min, whereas endocytosis of the ΔGY CD construct was reduced to 1.8 ± 0.6% per min, a rate consistent with bulk

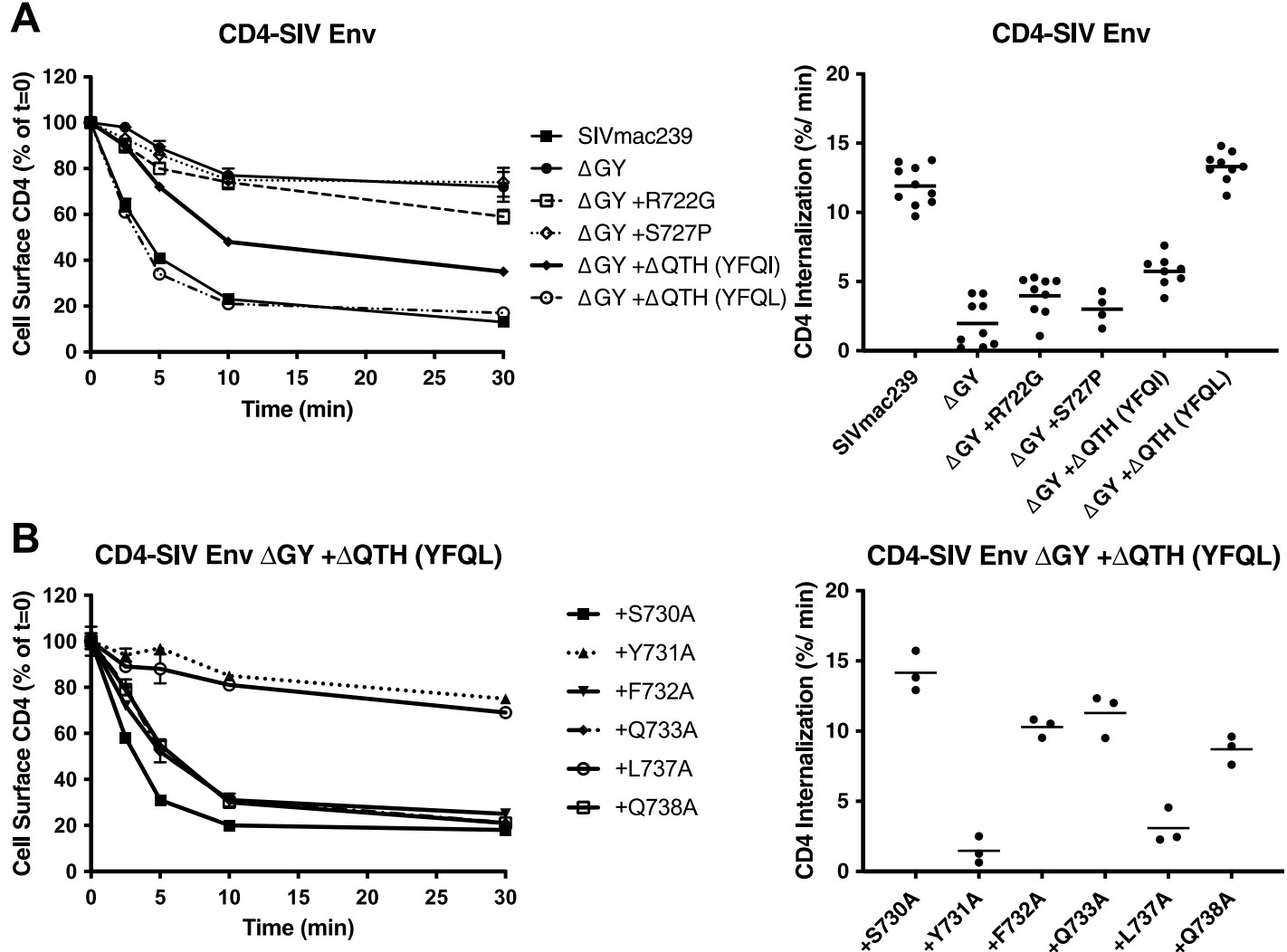

**Fig 4. ΔGY and *in vivo* acquired a.a changes modulate endocytic rates.** (**A**) Endocytosis of CD4-SIV Env CD chimeras shown in Fig 3A in HeLa cells. (**B**) Endocytosis of CD4-SIV Env CD constructs containing ΔGY + ΔQTH (YFQL) with the indicated Ala substitutions (SIVmac239 a.a. numbering). Left panels show cell surface CD4 for the indicted times as a % of 0 mins (= 100%); right panels show the rate of endocytosis over the first 5 mins after warming up. Graphs display the mean ± S.E.M. (left) or mean and individual values (right) from ≥ 3 independent experiments.

membrane turnover (Fig 4A). The R722G or S727P substitutions had little effect on internalization, whereas the ΔQTH deletions that generated YFQI or YFQL restored endocytic rates to 5.7 ± 0.4% and 12.8 ± 0.6% per min, respectively, the latter being similar to SIVmac239 CD constructs (Fig 4A).

To determine if the YFQL signal conformed to a conventional YxxØ motif, Ala substitutions were introduced into the ΔGY+ΔQTH (YFQL) construct, and the endocytosis rates determined on stably transformed HeLa cells (Fig 4B). Substitution of Y731 (position Y+0) or L737 (position Y+3) reduced endocytosis rates to background levels comparable to constructs bearing the ΔGY deletion alone, whereas substitutions at the Y+1, +2 or +4 positions had only minor effects, consistent with YFQL being a classical YxxØ-type endocytic signal [36]. Y731A also altered the distribution of a CD4-SIV Env CD construct from intracellular sites to the cell surface (S2A and S2B Fig). Moreover, depletion of the AP2 μ2 subunit, which is critical for the AP2 complex stability required for clathrin-mediated endocytosis, ablated endocytosis of ΔGY-CD constructs containing ΔQTH deletions (S3 Fig).

Thus, both ΔQTH deletions that occurred in SIVmac239ΔGY-infected macaques that progressed to AIDS, creating either YFQL or YFQI, generated highly efficient endocytosis signals similar to the parental SIVmac239 sequence (YRPV) and were confirmed to be Tyr- and AP2-dependent YxxØ signals.

## ΔQTH deletions also create new basolateral sorting signals

For HIV and SIV Envs, the Tyr in the membrane proximal GYxxØ motif has been shown to mediate basolateral (BL) sorting of Env expressed in polarized epithelial cells [25,27,28,37]. To determine if the YFQI and YFQL motifs generated by the ΔQTH deletions also reconstituted BL sorting, CD4-SIV Env CD constructs with or without these changes (Fig 3A) were stably expressed in MDCKII cells that polarize to form apical and BL surfaces when cultured as monolayers [38]. The panel of constructs included CD4-SIV Env CD chimeras from SIVmac239 with truncated or full-length CDs (Fig 5A and 5B) as well as full length SIVmac239 Env (Fig 5C). Surface expression of CD4-SIV Env constructs containing either SIVmac239 truncated or full-length CDs was low but localized to BL membranes as visualized by immunofluorescence microscopy and quantified by determining the BL/apical distribution ratio (Fig 5A and 5B). Introduction of Y721I or ΔGY resulted in complete loss of polarized sorting (Fig 5A and 5B), consistent with previous findings that BL sorting of HIV-1 and SIV Envs is dependent on this Tyr [25,27]. In contrast to the presence of multiple endocytic signals in Env CDs [22], these results indicate that polarized sorting of CD4-SIV Env CD chimeras was determined solely by the GYRPV motif. In agreement with these observations, a BL distribution was also seen for native, full-length SIVmac239 Env, which was ablated by a ΔGY deletion (Fig 5C). R722G or S727P substitutions in the CD4-SIV ΔGY Env truncated CD construct failed to restore polarized sorting (Fig 5A). However, when the ΔQTH deletions were created, introducing YFQI or YFQL sequences, BL sorting was fully restored to levels comparable to the SIVmac239 CD construct. A Y731A substitution in the YFQL sequence completely ablated this distribution (Fig 5A). Thus, the YxxØ motifs created by the ΔQTH deletions completely restored polarized sorting of CD4-SIV ΔGY Env CD constructs.

## Evaluating the effects of R722G and ΔQTH mutations *in vivo*

We sought to determine the impact of the R722G substitution and ΔQTH deletions on SIVmac239ΔGY pathogenesis *in vivo*. We selected PTM for this evaluation given the usually potent viral control and absence of disease in this species following SIVmac239ΔGY infection [18]. We first determined whether SIVmac239ΔGY viruses containing R772G, with or without

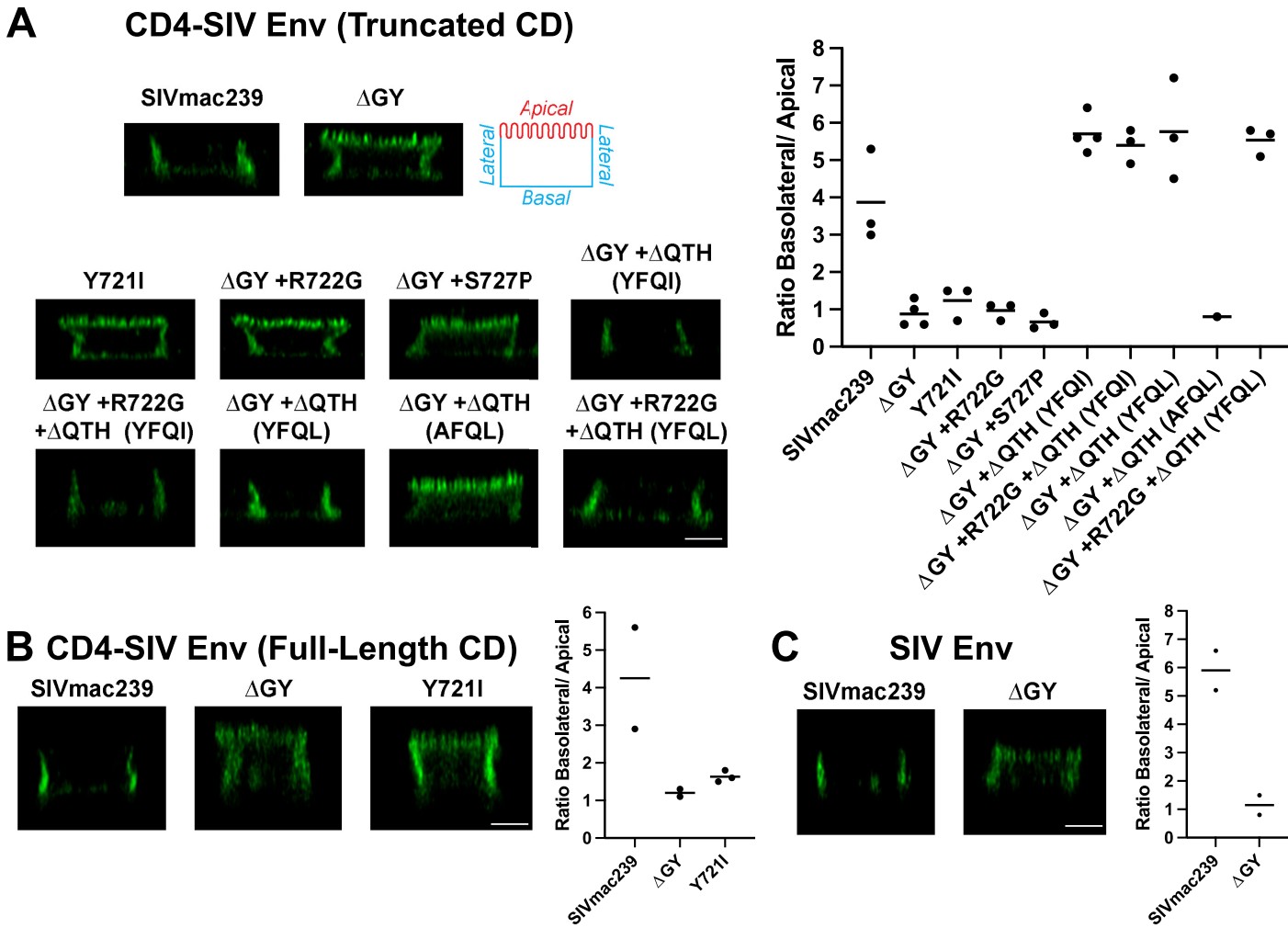

**Fig 5. ΔGY and *in vivo* acquired mutations modulate Env sorting in polarized epithelial cells. (A)** Cell surface distribution of CD4-SIV Env truncated CD chimeras containing the indicated Env CDs from Fig 3 on polarized MDCKII cells. Apical, basal, and lateral surfaces are indicated. **(B)** Cell surface distribution of CD4-SIV Env chimeras containing a full-length SIVmac239 CD. **(C)** Cell surface distribution of native, full length SIVmac239 Env with or without the ΔGY mutation. Left panels show orthogonal deconvolved immunofluorescence projections of representative cells; right panels show quantitation of the images as a ratio of the basal + lateral to apical labelling intensity calculated as described in the Material and Methods. Data shown are the means of measurements from 58–223 cells per condition imaged from **(A)** n≥3 (except ΔGY + ΔQTH [AFQL] which is n = 1) **(B)** n≥2 and **(C)** n = 2 independent experiments. Scale bar = 5 μm.

the ΔQTH deletions, were replication competent *in vitro* in PTM PBMCs (S4 Fig). While ΔGY-containing viruses with both R722G and the ΔQTH deletion, generating YFQL (SIVmac239ΔGY+R722G+ΔQTH), or R722G alone (SIVmac239ΔGY+R722G) replicated in PBMCs, those with ΔQTH alone replicated poorly. Therefore, we selected SIVmac239ΔGY +R722G+ΔQTH and SIVmac239ΔGY+R722G viruses for *in vivo* studies. Two groups of 3 PTM were inoculated i.v. with 300 $TCID_{50}$ of each virus and the animals followed for plasma viremia, CD4+ T cells in blood and gut, and the stability of mutations over time [32,33].

## Infection by SIVmac239ΔGY+R722G+ΔQTH resulted in persistent viremia

The 3 PTMs inoculated with SIVmac239ΔGY+R722G+ΔQTH (KV74, KV52, and KV76) exhibited acute peak plasma viral loads ranging from $4.6–6.0x10^6$ copies/ml (Fig 6A). Although

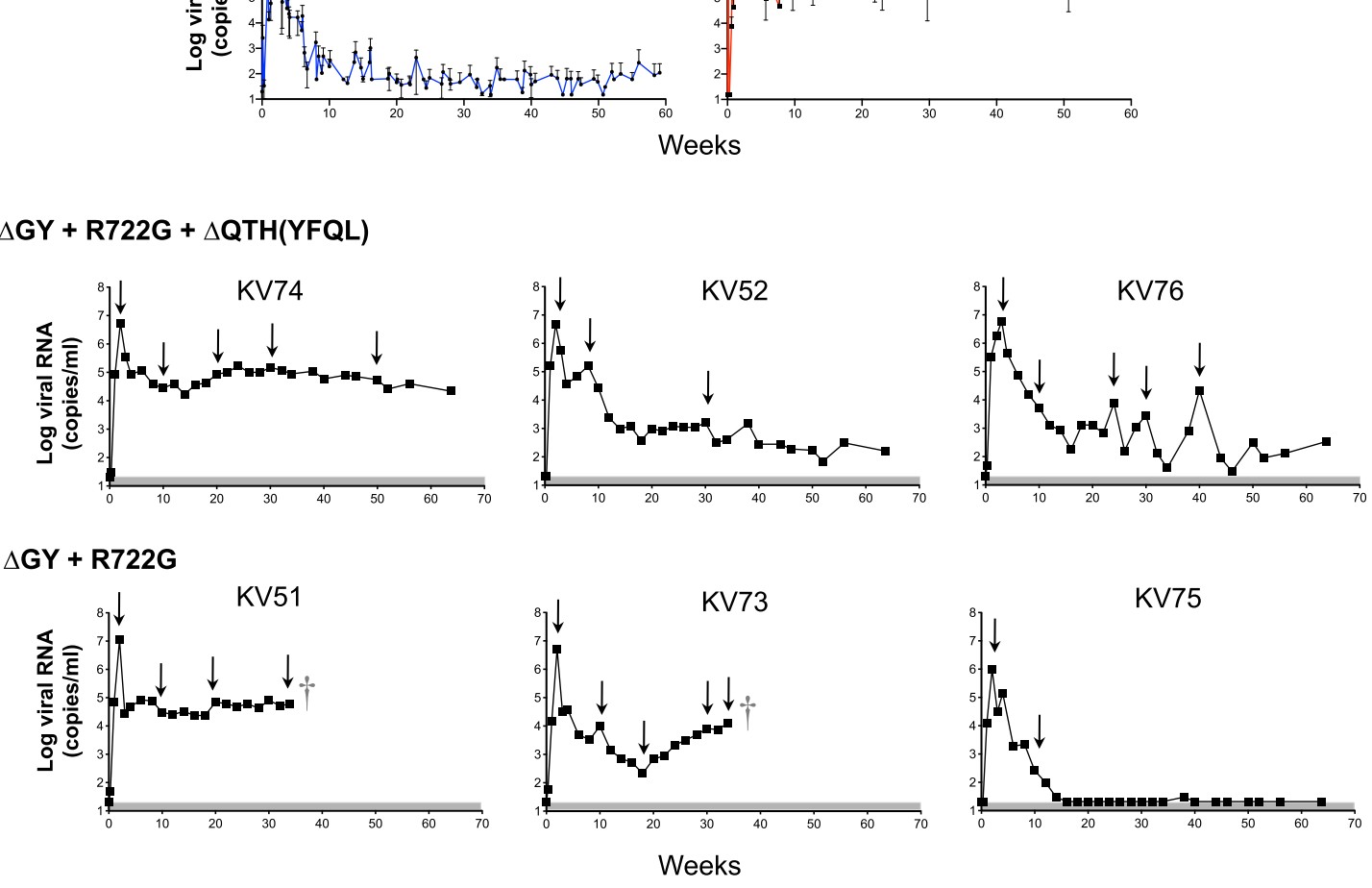

**Fig 6. Plasma viral loads for PTM infected with SIVmac239ΔGY containing mutations acquired *in vivo*.** Top panels show plasma viral loads from historical PTM controlling SIVmac239ΔGY (n = 23; left panel) and SIVmac239-infected controls (n = 14; right panel) (18). Lower panels show plasma viral loads for individual animals inoculated with SIVmac239ΔGY+R722G+ΔQTH (animals KV52, KV74 and KV76) and SIVmac239ΔGY+R722G (animals KV51, KV73 and KV75). † Indicates death due to an AIDS-related complication. Arrows denote time points at which plasma was obtained for SGS. Shaded areas indicate approximate. limits of assay sensitivity, which was 21 copies/ml for animals shown.

the levels of viral RNA varied during chronic infection, all animals maintained detectable viremia for 65 weeks with KV74 having high and stable levels ($0.4 \times 10^5$–$1.7 \times 10^5$ copies/ml), KV52 showing a gradual decline, and KV76 showing marked fluctuations (levels ranging from $0.3 \times 10^4$–$2.2 \times 10^4$ RNA copies/ml). Gut CD4+ T cells declined for all animals during the first 2–4 weeks of infection to levels that were lower than for historical ΔGY-infected PTMs, but then recovered with KV74 remaining at ~50% of pre-infection levels, and KV76 and KV52 showing a gradual return to baseline (S5 Fig). KV74 also showed a reduction in platelets, a recognized correlate of AIDS in PTM [39,40], between 16 and 40 weeks post-infection in combination with low levels of CD4+ T cells comparable to SIVmac239 (S5 and S6 Figs), although platelets in this animal subsequently recovered to baseline (S6 Fig).

SGS was performed at multiple time points (Fig 6). The sequences for the Env CD are shown for all amplicons (S7A–7C Fig) with a summary of changes shown for Env a.a. 719–752

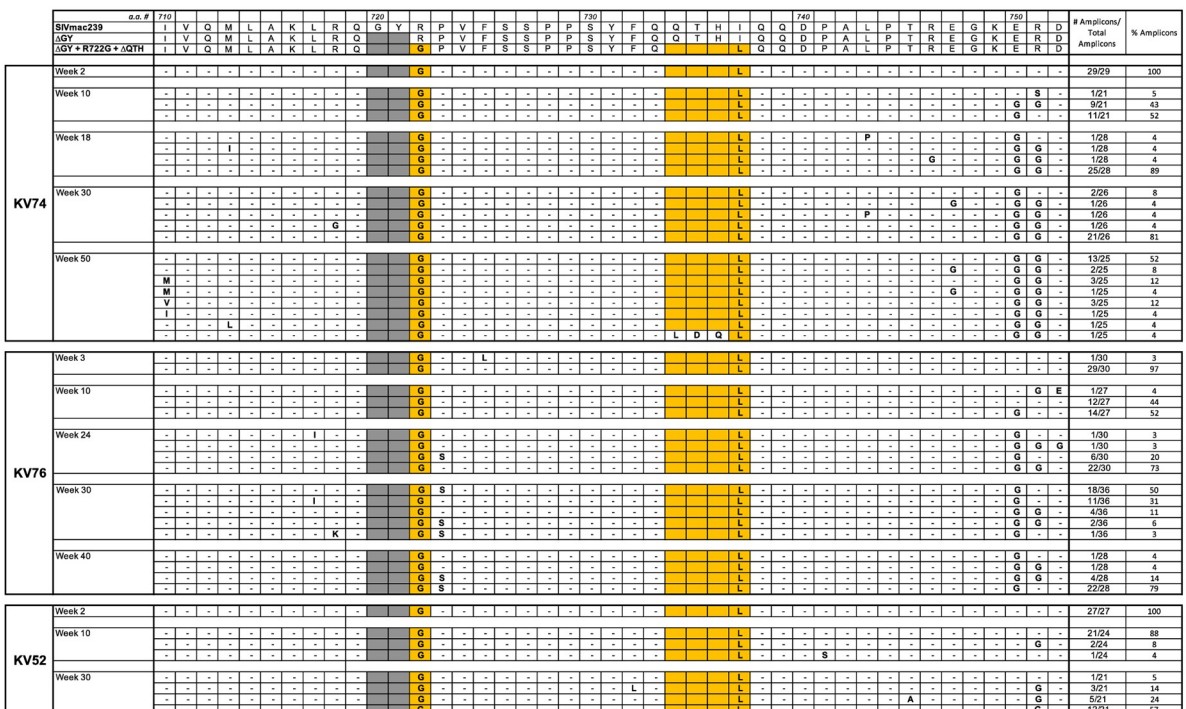

**Fig 7. Viral evolution in PTM inoculated with SIVmac239ΔGY+R722G+ΔQTH.** Summary of SGS from 3 PTM inoculated with SIVmac239ΔGY+R722G+ΔQTH. Results are shown for mutations within Env a.a. 710–752. The number and % of amplicons bearing the indicated mutations are shown. Dashes indicate identity with SIVmac239ΔGY+R722G+ΔQTH; shaded areas indicate a ΔGY deletion. Sequences in the top panel show Env of SIVmac239, ΔGY and ΔGY+R722G+ΔQTH from parent viruses. Colored boxes show a.a. changes introduced into the SIVmac239ΔGY background and their conservation over time in each animal. An I737L substitution was acquired adjacent to the ΔQTH deletion. The Env CD sequences of individual amplicons is given in S7 Fig.

covering regions for ΔGY, R722G, and ΔQTH (Fig 7) and Env a.a. 860–880 covering the distal CD (S1 Table). SGS showed persistence of the R722G substitution and ΔQTH deletion in all amplicons and at all time points except for 1 of 25 amplicons in KV74 at week 50 in which ΔQTH was lost (Fig 7 and S7A–7C Fig). Persistence of the ΔQTH deletion was remarkable given that this mutation creates deletions within the *tat* and *rev* second exons (S1 Fig). An analysis of *tat* and *rev* transcripts in PBMCs from all 3 animals assessed 14 days post infection confirmed that these deletions were present *in vivo* (S8 Fig).

A well described fitness mutation for parental SIVmac239 (R751G) [41], appeared in all animals by week 10, with or without an adjacent E750G substitution (S7A–7C Fig). Although less commonly observed *in vivo*, E750G has been reported in SIVmac239-infected macaques [42]. In animal KV76, a P723S substitution, not seen in other animals, appeared at week 24 and was nearly fixed by week 34 (Fig 7 and S7B Fig). Additional sporadic changes were seen in these animals including V815A and V837A and, in the C-terminus, G873E or R, L874I, and L879S, although none of these became fixed (S1 Table and S7A–7C Fig). Thus, the addition of R722G and ΔQTH to SIVmac239ΔGY resulted in a sustained high viremia in one animal and variable but persistent levels of viremia in two others for over 1 year. These findings are in marked contrast to SIVmac239ΔGY infection where viral control typically occurs within 8–10 weeks (Fig 6) [18]. Importantly, complete retention of R722G and ΔQTH indicated that there was strong selection pressure to maintain these mutations and their acquired functions *in vivo*.

## Infection with SIVmac239ΔGY+R722G leads to disease progression in association with new changes in Env CD

The 3 animals inoculated with SIVmac239ΔGY+R722G (KV51, KV73 and KV75) exhibited acute plasma viral peaks at week 2 of $1.1 \times 10^7$, $5.2 \times 10^6$, and $9.8 \times 10^5$ RNA copies/ml, respectively (Fig 6). In KV75, the viral load decreased to 100 copies/ml by week 12 and thereafter became undetectable. In this animal, gut CD4+ T cells decreased from 50% to 18% of T cells at week 2 but then increased to pre-infection levels as the viremia declined (S5 Fig). In contrast, KV51 poorly controlled the virus with viremia persisting between $2.3–8.5 \times 10^4$ copies/ml, while in KV73, viremia decreased to a low of 210 copies/ml at week 18, but thereafter increased to $1.3 \times 10^4$ copies/ml by week 35. Gut CD4+ T cells for both animals decreased to <5% at the time of peak viremia, with KV51 remaining at 25% of pre-infection levels and KV73 returning to its pre-infection level by week 28 (S5 Fig). Notably, KV51 and KV73 developed severe thrombocytopenia (S6 Fig) and died at weeks 36 and 37, respectively, with massive pulmonary artery thrombi, a complication frequently seen in PTM during pathogenic SIV infection [18,43–46].

SGS of plasma virus from KV75, at weeks 2 and 10, showed that the ΔGY deletion and R722G substitution were maintained (Fig 9 and S7D Fig) and at week 10 both R751G and E750G appeared. Only one additional change in Env appeared, an A836D substitution in 19 of 20 amplicons (S7D Fig and S2 Table) within a CTL epitope targeted in both RM [47] and PTM [48]. In marked contrast, KV51 and KV73 exhibited striking new changes during progression to disease. In KV51, the ΔGY and R722G changes were maintained, but at week 24 an in-frame 9 nt deletion (nt 8803–8011) appeared in 26 of 29 amplicons (Fig 9 and S7E Fig) that generated a QTH deletion with a new YFQI sequence (Fig 1 and S1 and S8 Figs). In 7 amplicons, a YFQL sequence resulting from an I737L substitution, was also found. By week 34, all 26 amplicons contained ΔQTH, 24 with YFQL and 2 with YFQI, indicating a selection advantage for YFQL (Fig 9). Distal to the R751G substitution with or without an adjacent E750G, no other changes were seen in >10% amplicons except for an L874I substitution near the Env C-terminus at week 34 (S2 Table and S7E Fig).

In KV73, ΔGY and R722G were also conserved throughout infection (Fig 9 and S7F Fig). Interestingly, a ΔQTH mutation appeared in 1 of 22 amplicons at week 10 but was lost at all subsequent time points. However, starting at week 10, a.a. substitutions were observed between positions 735 and 744 of Env (Fig 9) including T735I, H736Y, Q739R, P741Q or L, and P744L. The proportions of these changes varied but evolved by week 34 to a consensus of T735I, Q739R, and P744L in 20 of 21 amplicons (henceforth termed the "IRL set"; Fig 9). Analysis of publicly available sequences indicated that T735I and Q739R have been observed individually during SIVmac239 infection but never together, while P744L has not been reported. As shown (S9 Fig), nt mutations that created the T735I and H736Y in Env, along with a G to A nt mutation that is silent in Env, generated 3 coding mutations in the *tat* second exon. Notably, the mutation responsible for Env P744L created a stop codon that deleted the last 22 a.a. of Tat. This truncation was verified by RT-PCR and sequencing of viral transcripts when a virus containing the IRL set was grown *in vitro* in PTM PBMCs. The mutations generating V837A and G873E also appeared at week 10 and became fixed (S2 Table and S7F Fig).

Thus, the two animals infected with SIVmac239ΔGY+R722G that progressed to AIDS evolved new changes in the Env CD: in KV51, a ΔQTH deletion generating YFQL, and in KV73, the IRL set. The previous appearance of ΔQTH in RM and now in PTM again suggested strong selection pressure to restore trafficking signals for Env endocytosis and polarized sorting. However, the IRL set did not correspond to any known cellular trafficking signal.

### The IRL set confers a novel signal for Env polarized sorting but not endocytosis

To determine if the IRL set acquired in animal KV73 during progression to AIDS influenced Env trafficking, substitutions, individually and in combination, were introduced into CD4-SIV Env CD chimeras containing ΔGY+R722G (Fig 8A). Constructs containing the

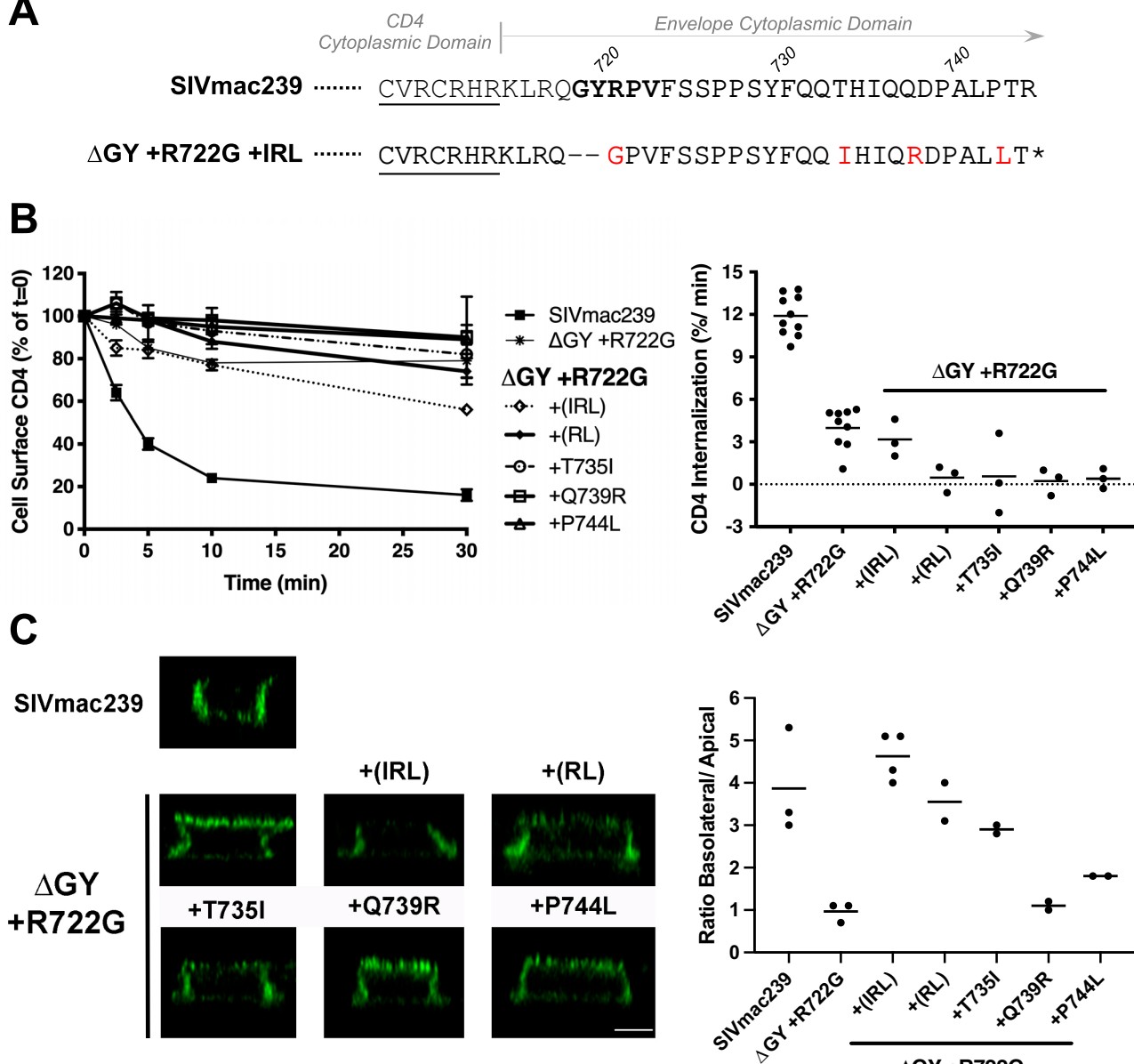

**Fig 8. The IRL set restores polarized sorting of Env but not endocytosis. (A)** The C-terminal sequences of CD4-SIV Env CD constructs (Fig 3A) with sequences from the CD4 CD underlined. Constructs contained the CD from WT SIVmac239 or SIVmac239ΔGY+R722G+IRL (red). **(B)** Endocytosis of CD4-SIV Env truncated CD constructs in HeLa cells. Left panel shows the % of cell surface CD4 at the indicted times (0 mins = 100%); right panel shows the rate of CD4 endocytosis over the first 5 mins after warm-up. Graphs display the mean ± SEM (left panel) or the values from 3 independent experiments (right panel). **(C)** Cell surface CD4-SIV Env CD constructs on polarized MDCKII cells. Left panels show orthogonal deconvolved immunofluorescence projections of representative cells; right panel shows quantitation of the ratio of basolateral to apical signal from the left panels. The data show the mean of between 59–190 cells per condition and the values from n≥2 independent experiments. Scale bar = 5 µm.

IRL set or single amino acid changes within IRL (T735I, Q739R, or P744L) showed no change in endocytosis rates compared to ΔGY and displayed minimal internalization of anti-CD4 antibody (Figs 8B and S2). In contrast, chimeras containing all 3 IRL substitutions showed potent BL sorting in MDCKII cells, equivalent to that of SIVmac239 Env CD (Fig 8C). Individual T735I and P744 substitutions partially reconstituted BL sorting, although all 3 were required for maximal effect. These results indicated that during SIV infection *in vivo* there is strong selection pressure for polarized trafficking of Env. Moreover, the finding that, in contrast to ΔQTH, IRL restored polarized sorting but not endocytosis suggests that, at least for the membrane proximal GYxxØ motif, polarized trafficking of Env is the dominant function *in vivo*.

## The polarized sorting function of the IRL set is conserved *in vivo* and confers persistent elevated levels of viremia to SIVmac239ΔGY

We next determined if the IRL set was sufficient to restore pathogenicity to a virus containing ΔGY. Given the absolute conservation of R722G in animal KV73 that developed the IRL set, a SIVmac239ΔGY+R722G+IRL virus was produced, shown to be replication competent in PTM PBMCs, and inoculated i.v. into 4 PTM. These animals (NH85, NH86, NH87 and NH88) exhibited intermediate to high acute viral RNA peaks ($2.3 \times 10^7$, $3.4 \times 10^7$, $2.7 \times 10^5$, and $1.7 \times 10^6$ copies/ml, respectively) and maintained elevated plasma viremia ($1.3 \times 10^3$, $2.5 \times 10^3$, $3.8 \times 10^4$, and $5.1 \times 10^5$ copies/ml, respectively) for up to 30 weeks (S10 Fig). Thereafter, NH85 and NH86 decreased to <100 copies/ml, while NH87 and NH88 increased to terminal values at week 40 of $9.2 \times 10^4$ and $2.3 \times 10^5$ RNA copies/ml, respectively. Gut CD4+ T cells transiently decreased in 3 animals but recovered to pre-infection levels (S5 Fig). The persistent viremia in these animals was in marked contrast to historical PTM inoculated with SIVmac239ΔGY, which exhibited viral set points typically <15–50 copies/ml by 10–20 weeks of infection [18] (see also Fig 6).

SGS of plasma virus was performed at multiple time points (S10 Fig), and results for the Env CD are shown for all amplicons (S11 Fig) with a summary of changes shown for a.a. 710–757 encompassing ΔGY, R722G, and the IRL set (S3 Table), and a.a. 800–880 encompassing the distal CD (S4 Table). In 2 animals (NH85 and NH87) amplicons containing a G873E near the Env C terminus appeared at week 12 and included all amplicons by week 33 (S4 Table and S11A and S11C Fig). Interestingly, G873E also become a consensus substitution in KV73, the animal that initially developed the IRL set (S2 Table and S7F Fig), and in a minority of amplicons in 2 of 3 animals infected with SIVmac239ΔGY+R722G+ΔQTH (S1 Table and S7 Fig). However, a G873R at this position appeared in NH86 and NH88 (in 100% and 93% of amplicons, respectively) suggesting that loss of G873 was likely driven by immune pressure rather than acquisition of a new trafficking signal. Additional mutations appeared in the CD, but none were common to all animals. As expected, all 4 animals developed R751G by week 12, consistent with ongoing replication of a SIVmac239-based virus [41]. Importantly, SGS from weeks 2, 12 and 33 post infection revealed conservation of all 3 IRL substitutions in nearly every amplicon at each time point, including P744L, which, as noted previously, generated a premature stop codon in *tat* (S3 Table and S9 Fig).

Collectively, these findings indicate that the novel IRL polarized sorting signal acquired during pathogenic evolution of a SIVmac239ΔGY+R722G virus *in vivo*, was completely conserved in *de novo* infections of naïve PTM. Although not sufficient to confer AIDS, at least through 33 weeks of infection, these findings indicated that acquisition of this novel signal for polarized sorting, but not endocytosis, was sufficient to restore high replicative capacity and fitness to a SIVmac239ΔGY+R722G virus.

## Discussion

The cytoplasmic domain of HIV and SIV Env contain a highly conserved Tyr-based trafficking motif (GYxxØ) that mediates both clathrin-dependent endocytosis [17,20–22] and polarized sorting [27,37,49]. Though less studied, similar motifs have been described in the Env CD of other retroviruses, including HTLV-1 [16]. For HIV and SIV Env, the GYxxØ motif can mediate endocytosis through interaction with the clathrin adaptor protein AP2, as seen for cellular proteins that contain similar Tyr-based YxxØ signals [17,20–22,50]. By contrast, the pathways and cellular partners required for polarized sorting of HIV and SIV Env are less well defined, but the Tyr within the GYxxØ motif is critical [25]. Additional motifs including di-leucine, [D/E]xxxL[L/I], DxxLL and acidic clusters, have also been implicated in both the endocytosis and polarized sorting of cellular proteins [36] and HIV and SIV Env [22,51]. Although there are examples of how the loss of viral Env trafficking motifs can alter pathogenesis in small animal models [52,53], a role of these signals in HIV and SIV infection *in vivo* has been unclear. Here we provide the first demonstration that the membrane proximal GYxxØ in the SIV Env CD is not only crucial for SIV pathogenesis in macaques, but that the individual functions associated with this motif are under strong positive selection and that loss of these functions can lead to potent host immune control reminiscent of elite HIV control in humans.

Several pathogenic roles have been proposed for trafficking functions in HIV and SIV Envs. Because Env delivered to the cell surface is rapidly internalized, resulting in low steady state levels of Env on the plasma membrane of infected cells, we and others have proposed that Env endocytosis renders cells less susceptible to antibody attack either directly or by antibody-dependent cell-mediated cytotoxicity [21,54,55]. Consistent with an *in vivo* role for the SIV-mac GYRPV motif, SIVmac239 with a T721I substitution remained stable during extensive serial passaging *in vitro*, but rapidly reverted in both RM and PTM [34]. In contrast, the ΔGY mutation in SIVmac239 reduced Env content on cells and virions (Fig 2), and a virus containing ΔGY was controlled in PTMs. For HIV, substitutions of the Tyr within the analogous GYSPL consensus sequence are poorly tolerated even during *in vitro* replication [56], suggesting a role for this motif in assembly and/or infectivity, although this effect can vary with different viral strains [56,57]. In contrast to endocytic function, the relevance of polarized sorting of HIV and SIV Env *in vivo* has been unclear, although this property has been recognized to positively affect viral infection and cell-cell spread *in vitro* [25–28].

We have shown that the ΔGY deletion in SIVmac239 Env CD, results in a novel phenotype *in vivo*, in which a majority of PTM suppress viral replication through cellular immune responses but not neutralizing antibodies [18]. Despite robust replication in lymphoid tissues, there is only transient infection of gut CD4+ T cells, no detectable infection of macrophages, and little to no immune activation [18]. Nonetheless, progression to AIDS has been reported in SIVmac239ΔGY-infected RM [32] and PTM [18] in association with novel changes in the Env CD. However, the role of these changes and the extent to which they compensate for defects introduced by the ΔGY deletion have been unclear [58].

In this study we evaluated mutations acquired during pathogenic SIVmac239ΔGY infection, including an R722G substitution flanking the ΔGY deletion and the loss of 3 amino acids (ΔQTH) resulting from 9 nt deletions within overlapping reading frames for the 2nd exons of *rev* and *tat*. The ΔQTH deletions created novel YFQI or YFQL sequences in Env (Fig 1) that restored both the endocytic and polarized sorting functions of the parental GYRPV motif. These functions depended on the Tyr, and for endocytosis, required AP2, indicating that *bone fide* cellular trafficking signals had been reestablished that correlated with progression to AIDS. We also showed that the ΔGY deletion reduced Env on virions, likely due to a general reduction in Env content in infected cells (Fig 2), and that this defect could be rescued by

R722G but not ΔQTH. While R722G did not restore endocytosis or polarized sorting, it was critical for maintaining replication fitness of ΔGY viruses containing ΔQTH deletions (S4 Fig). We also demonstrated that an S727P substitution, previously seen in SIVmac239ΔGY infected RM [33] and one PTM [34] that progressed to AIDS, also restored Env content on ΔGY infected cells to near wildtype levels (Fig 2), similar to the R722G substitution, though its effect on Env levels in virions was modest. In all ΔGY infected animals that progressed to disease either R722G or S727P appeared, indicating that restoring Env expression levels was critical.

To examine the role of these novel changes *in vivo*, we infected PTM with SIVmac239ΔGY containing R722G and ΔQTH. Notably, the mutations encoding these amino acid changes were retained at all time points (S7 Fig). In contrast to SIVmac239ΔGY infected PTM, where viral loads are typically controlled to low or undetectable levels [18], partial reconstitution of SIVmac239 pathogenicity was observed, with one animal (KV74) progressing to AIDS and detectable viremia persisting in all 3 animals for up to 64 weeks (Fig 6). When PTM were infected with SIVmac239ΔGY containing R722G alone, 2 of 3 animals progressed to AIDS with high viral loads that were associated with the appearance of additional changes in the Env CD, either 1) a new ΔQTH deletion generating a YFQL (Fig 9 and S7E Fig), or 2) 3 substitutions (T735I, Q739R, and P744L) that evolved to a consensus sequence (Fig 9 and S6F Fig). This latter 'IRL set' generated a novel signal for polarized sorting, but not endocytosis. Remarkably, the mutation encoding P744L created a premature stop codon within the *tat* 2nd exon. When a SIVmac239ΔGY virus containing R722G and the IRL substitutions was given to 4 PTM, all animals maintained persistent viremia to 30 weeks and the IRL set was highly conserved in all animals through 33 weeks of infection.

Our findings that a ΔQTH deletion or the IRL set, which restored both Env endocytosis and polarized sorting (for ΔQTH) or polarized sorting alone (for the IRL set), were retained

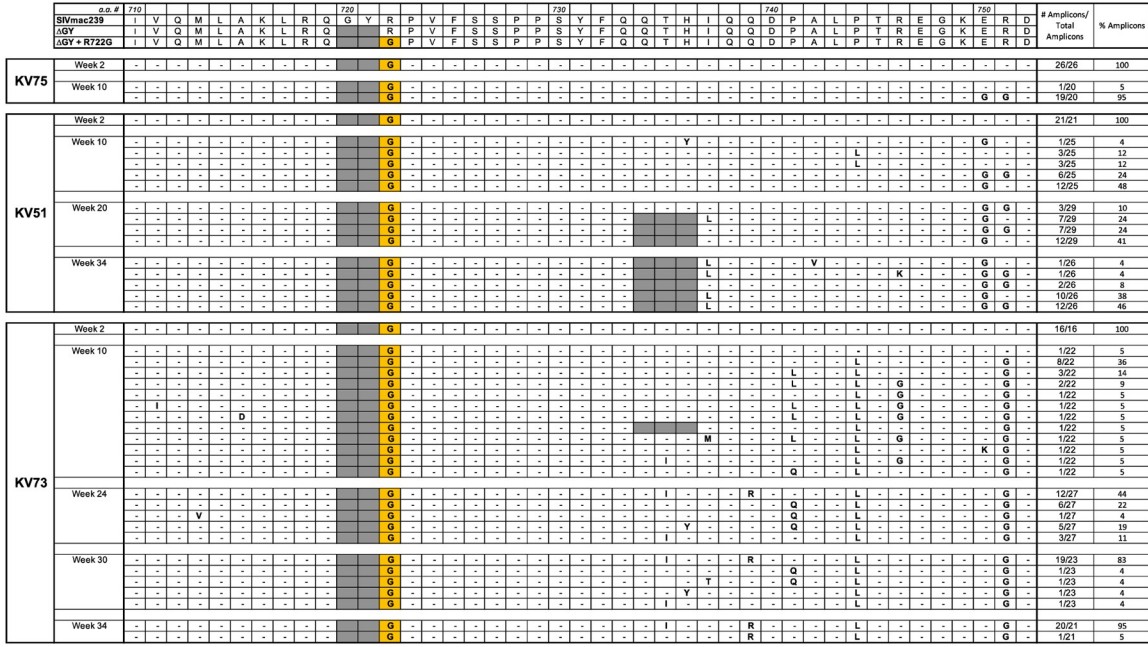

**Fig 9. Viral evolution in PTM inoculated with SIVmac239ΔGY+R722G.** Summary of SGS of plasma virus performed at the indicated time points for 3 PTM inoculated i.v. with SIVmac239ΔGY+R722G. Results are shown for changes within a.a. 710–752 as in Fig 7. Sequences in the top panel show Env of SIVmac239, SIVmac239ΔGY and SIVmac239ΔGY+R722G. Complete CD a.a. sequences for individual amplicons are shown in S7 Fig.

during *de novo* infections, indicate that these trafficking functions are likely critical for pathogenic SIV infection. While AP2-mediated Env endocytosis can be directed by the membrane proximal GYxxØ motif and membrane distal LL-containing motifs [21,22,51,59,60], experiments with truncated and full-length CD constructs in MDCK cells indicated that the GYxxØ motif is the only determinant in the SIV Env CD mediating polarized sorting (Fig 5). The finding that the IRL set restored the BL sorting function lost by the ΔGY deletion, but not endocytosis (Fig 8 and S2 Fig) and was sufficient to confer persistent viremia to a SIV-mac239ΔGY+R722G virus, suggests that polarized sorting of Env may be the principal trafficking function of the GYxxØ motif during progression to disease. This is not to say that Env endocytosis is not important, rather that this function may also be carried out by less well-defined endocytic signals present in SIV Env CD [22], similar to the highly conserved C-terminal di-leucine that we have shown mediates HIV Env endocytosis [51]. Nevertheless, for the 4 animals given the SIVmac239ΔGY+R722G+IRL virus, addition of this polarized sorting signal imparted higher and more sustained viral loads.

The mutations generating ΔQTH and IRL arose in an area of the genome in which all three reading frames are used and have the potential to alter transcripts for Tat and Rev. Indeed, changes in these genes were demonstrated and included, for ΔQTH, deletions and loss of a splice acceptor site (S8 Fig), and for IRL, point mutations and premature truncation of the *tat* 2nd exon (S7 Fig). Although Tat truncations within this exon can be tolerated for SIV and HIV *in vitro* [61,62], in RM infected with SIVmac239 lacking a *tat* 2nd exon, reversion to a two exon *tat* occurred in association with high viral loads and falling CD4+ T cell counts, while persistence of a single exon *tat* was associated with viral control [63]. It is therefore remarkable that nt changes that generated P744L during evolution of the IRL set also created a premature stop codon in *tat* that was maintained when SIVmac239ΔGY containing these mutations was used to infect naïve animals. Thus, the IRL signal for polarized Env trafficking was maintained at the expense of the *tat* 2nd exon further emphasizing the strong selection pressure *in vivo* to generate and maintain this Env trafficking function.

Unlike epithelial cells, which maintain fixed apical and BL plasma membrane domains, T cells undergo polarization during their migration along chemokine gradients [64–68], during the formation of immunological synapses with antigen presenting cells or the cellular targets of cytotoxic T cells [69], and when virological synapses (VS) form between virally infected and uninfected cells [70]. For HIV, VS require interactions between Env and CD4 [71–75] to enhance the efficiency of infection and cell-cell spread *in vitro* [25,73,76,77]. Although HIV Gag and RNA colocalize at the uropod of polarized T cells in an Env-independent manner [29,31,78], Env is also present at this site, indicating that, like murine leukemia virus [79] and measles virus [52], the uropod is a site for viral assembly as well as engaging CD4 on target cells to nucleate synapse formation [73,76]. It is likely that polarized trafficking of Env contributes to this process. While VS and polarized sorting of Env have been recognized as contributing to pathogenesis of MuLV [53,79] and measles virus [52], there is little direct evidence that this function is important for HIV or SIV *in vivo*. Our findings that ΔQTH and the IRL set, acquired in SIVmac239ΔGY-infected animals that progressed to AIDS, regenerated cellular trafficking signals for endocytosis and/or polarized sorting and that they were retained during *de novo* infections, indicates that these trafficking functions are likely critical for SIV replication and pathogenesis.

In summary, our characterization of pathological revertants in SIVmac239ΔGY infected macaques highlight critical *in vivo* roles played by cellular trafficking motifs in the SIV and, by analogy, HIV Env CDs. Whereas the ΔGY deletion within the conserved GYxxØ motif ablates endocytosis and polarized sorting, these functions were regained through novel deletions or substitutions at the expense of collateral changes in Tat and Rev. Given that SIVmac239ΔGY

replicates poorly in gut CD4+ T cells and fails to infect macrophages *in vivo* [18,32], our findings suggest that trafficking functions, particularly the polarized sorting of Env, could be required for optimal infection of these cells, perhaps by promoting VS formation and viral spreading through cell-cell contacts. VS enhance the efficiency of viral infection and cell-cell spread *in vitro* [25,76], at least in part, through the generation of transcriptional signals that enable viruses to overcome diverse barriers to infection, including restriction factors, neutralizing antibodies and reduced levels of receptors on target cells [73,77,80–83]. Studies are ongoing in SIVmac239 and SIVmac239ΔGY infected macaques to directly assess VS formation *in vivo* as well as the contributing role of compensatory mutations that may promote cell-to-cell spread.

## MATERIALS AND METHODS

### Ethics statement

Pigtail macaques used in this study were purpose bred at either the University of Washington National Primate Research Center or Johns Hopkins and moved to Tulane for these experiments. Macaques were housed in compliance with the NRC Guide for the Care and Use of Laboratory Animals and the Animal Welfare Act. Animal experiments were approved by the Institutional Animal Care and Use Committee of Tulane University (protocols P0088R, P0147, and P0312). The Tulane National Primate Research Center (TNPRC) is fully accredited by AAALAC International (Association for the Assessment and Accreditation of Laboratory Animal Care), Animal Welfare Assurance No. A3180-01. Animals were socially housed, indoors in climate-controlled conditions with a 12/12-light/dark cycle. All the animals on this study were monitored twice daily to ensure their welfare. Any abnormalities, including those of appetite, stool, behavior, were recorded and reported to a veterinarian. The animals were fed commercially prepared monkey chow twice daily. Supplemental foods were provided in the form of fruit, vegetables, and foraging treats as part of the TNPRC environmental enrichment program. Water was available at all times through an automatic watering system. The TNPRC environmental enrichment program is reviewed and approved by the IACUC semi-annually. Veterinarians at the TNPRC Division of Veterinary Medicine have established procedures to minimize pain and distress through several means. Monkeys were anesthetized with ketamine-HCl (10 mg/kg) or tiletamine/zolazepam (6 mg/kg) prior to all procedures. Preemptive and post procedural analgesia (buprenorphine 0.01 mg/kg or buprenorphine sustained-release 0.2 mg/kg SQ) was required for procedures that would likely cause more than momentary pain or distress in humans undergoing the same procedures. The above listed anesthetics and analgesics were used to minimize pain or distress associated with this study in accordance with the recommendations of the Weatherall Report. The animals were euthanized at the end of the study using methods consistent with recommendations of the American Veterinary Medical Association (AVMA) Panel on euthanasia and per the recommendations of the IACUC. Specifically, the animals were anesthetized with tiletamine/zolazepam (8 mg/kg IM) and given buprenorphine (0.01 mg/kg IM) followed by an overdose of pentobarbital sodium. Death was confirmed by auscultation of the heart and pupillary dilation. The TNPRC policy for early euthanasia/humane endpoint was included in the protocol in case those circumstances arose.

### Antibodies, Reagents and cell lines

*Antibodies*: The following reagents were obtained from the Centre for AIDS Reagents National Institute for Biological Standards and Control [NIBSC], South Mimms, UK): Anti-Gag p57/27 (SIV 27c donated by P. Szawlowski or KK60 donated by K. Kent), anti-Nef (KK77 donated by

Dr K. Kent), anti-CD4 (Q4120, provided by Q. Sattentau, University of Oxford). Murine anti-SIVmac gp120/gp160 monoclonal antibody DA6 and anti-SIVmac gp41 mAb 35C11 have been previously described [84]. Anti-Gag p27 (3A8 was provided by J. McClure). Anti-CD3 (SP34), CD4 (L200) and CD8 (SK1 or SK2) were used for flow cytometry and obtained from BD Biosciences.

*Reagents*: Human sCD4 domains 1–4 (ARP6000 Progenics Pharmaceuticals, Inc, USA was obtained from the Centre for AIDS Reagents, NIBSC) and human sCD4 domains 1–2 (#7356 from Pharmacia, Inc was obtained through the NIH AIDS Reagent Program, Division of AIDS, NIAID, NIH).

*Cell Lines*: Human embryonic kidney cells (HEK-293T; ATCC, CRL-3216), rhesus macaque kidney cells (LLC-MK2; ATCC, CCL-7), HeLa cells (from D. Cutler, MRC LMCB, UCL) and Madin Darby canine kidney cells (MDCKII from G. van Meer, University of Utrecht) were maintained in DMEM supplemented with 10% fetal calf serum (FCS; Sigma A7906) and 100 U/ml Pen/100 μg/ml Strep (Gibco, 15140–122); human T lymphoid cell lines CEMx174 and BC7 [85] a CD4 negative derivative of SupT1, were maintained in RPMI containing 10% FCS and 100 U/ml Pen/100 μg/ml Strep. All lines were monitored for mycoplasma infection using the MycoAlert Mycoplasma Detection Kit (Lonza TL07-218).

## VSV-G pseudotyping of SIVmac239 and virus titration

HEK293T cells were co-transfected with full-length SIVmac239 genome constructs and a plasmid encoding the VSV-G glycoprotein (pMD2.G; provided by P. Mlcochova, Div. Infection and Immunity, UCL), at a ratio of 3 μg of provirus to 1 μg of pMD2.G, using the Fugene6 transfection reagent (Promega). Virions were concentrated from the culture medium 48 hours post infection (hpi) by first clearing large debris (centrifugation for 5 min at 2000 rpm) followed by ultracentrifugation through a 20% (w/v) sucrose cushion (23,000 rpm [98,000 × g], 2 hr, 4°C). The pellet was suspended in culture medium (RPMI-1640, 100 U/ml Pen/100 μg/ml Strep and 10% FCS) and stored under vapor phase of liquid nitrogen. Infectious titres were measured by serial dilution on LLC-MK2 cells; following infection for 48 hr, cells were washed in PBS pH 7.4, fixed (3% [w/v] formaldehyde, PBS pH 7.4) for 30 min at 4°C and stored overnight (0.1% [w/v] formaldehyde, PBS pH 7.4). The following day, samples were quenched (50 mM NH$_4$Cl, PBS pH 7.4) for 15 min at room temperature (RT), permeabilized (0.1% [w/v] saponin, 1% [v/v] FCS, PBS pH 7.4) and immunolabeled with primary antibodies (anti-Gag [SIV 27c] and/or anti-Nef [KK77]) for 1.5 hr at RT followed by secondary antibodies conjugated to Alexa Fluor dyes (Invitrogen) for 1 hr at RT. Viral protein expression was detected by imaging with Opera LX and Phenix high content imaging platforms (Perkin Elmer) and the number of infected cells determined using a Columbus Analysis system (Perkin Elmer).

## Biochemical analysis of viral protein expression by western blotting

LLC-MK2 cells were incubated with VSV-G pseudotyped SIVmac239 to infect 40% of the cell population. The cell culture medium was replaced at 24 hpi and the cells cultured for a further 48 hrs. At 72 hpi, viruses in the culture supernatants were recovered by centrifugation through sucrose, as described above, and the corresponding cells were lysed in 150 mM NaCl, 1% (v/v) Triton, 50 mM Tris/HCl pH 8.0 containing complete protease inhibitors (Roche). The lysates were cleared of insoluble material and stored at -80°C prior to analysis. Cell and viral lysates were mixed with Laemmli Sample Buffer containing 100 mM dithiothreitol (DTT) and heated for 10 min at 98°C. To enable clear separation of Env and Gag proteins on the same gel, proteins were separated on Laemmli SDS-polyacrylamide gels where the resolving gel comprised an upper gel of 8% acrylamide over an equal volume of 15% acrylamide. Following

electrophoresis, the proteins were electroeluted from gels onto Immobilon-F PVDF membranes (Millipore) at 100 mA for 16–18 hr at 4˚C under wet blotting conditions (10 mM 3-[Cyclohexylamino]- 1-propanesulfonic acid pH 11.0, 10% [v/v] methanol). Membranes were blocked (5% [w/v] non-fat dried milk in 0.1% [v/v] Tween-20, PBS pH 7.4) for 3 h, followed by detection of proteins with primary antibodies (0.5% [v/v] Tween-20/ 1% [w/v] bovine serum albumin [BSA], PBS pH 7.4) for 1.5 hr at RT, or 4˚C overnight, followed by secondary antibodies conjugated to IRDye 800CW or IRDye 680 (LiCOR Biosciences) for 1.5 hr at RT. Viral proteins were detected with antibodies to gp120/gp160 (DA6), gp41 (35C11) and Gag p57/p27 (KK60). Images of the western blots and the intensity of protein bands were obtained and quantified using an Odyssey infrared imaging system (LiCOR Biosciences).

## Biochemical analysis of cell surface protein levels

At day 3 post infection, LLC-MK2 cells (40% infected) were washed with ice-cold PBS and cell surface proteins covalently labelled with cell impermeable EZ-Link Sulfo-NHS-S-S-Biotin (0.5 mg/ml; Pierce) for 45 mins at 4˚C. Excess label was removed and the samples quenched by washing with TBS (154 mM NaCl, 10 mM Tris/HCl pH 7.4) at 4˚C. Cell lysates were prepared as described above and diluted to equal protein concentrations. An aliquot of each lysate (150 μg of protein) was incubated with 100 μl, 50% slurry of NeutrAvidin Agarose beads (Pierce) overnight at 4˚C with inversion. To show that all the biotinylated proteins were captured with this first incubation, the lysate was separated from the beads and the process repeated with fresh NeutrAvidin beads for 3 hr. Subsequently, the beads were washed once with lysis buffer, once with TBS and once with TE (10 mM Tris/HCl pH 7.4 and 5 mM EDTA) and eluted twice by incubation with Laemmli sample buffer, containing 100 mM DTT, and heating for 10 min at 98˚C. Cell lysates ('L'; equivalent to 30 μg of protein) were separated alongside the proteins eluted from the NeutrAvidin beads ('S'; Surface) on SDS-PAGE gels, as described above. Following electrophoresis, the proteins were transferred to Immobilon-F PVDF membranes (Millipore) at 0.8 mA/cm$^2$ for 2 hr at RT under semi-dry blotting conditions using a discontinuous 3 buffer system (Anode Buffer I: 0.3 M Tris pH 10.4, 10% MeOH; Anode Buffer II: 25 mM Tris pH 10.4, 10% MeOH; Cathode Buffer: 25 mM Tris, 40 mM 6-amino-n-caproic acid pH 10.4) modified from [86]. Proteins were detected and imaged as described above.

## *In vitro* replication of SIV isolates in rhesus and pigtail macaque PBMCs

Purified peripheral blood mononuclear cells (PBMCs) from rhesus or pigtail macaques stored at -140˚C were thawed and cultured for 72 hours in RPMI with 5 μg/ml Concanavalin A (Sigma-Aldrich) at a concentration of 2–3 x 10$^6$ cells/ml. After 72 hr, cells were washed and resuspended 1x10$^6$ cells/ml in RPMI with 100 U/ml rHu IL-2 (Aldesleukin, Prometheus Laboratories, Inc.) and infected with viruses (250 ng of p27 Gag). After 24 hr, cells were washed and cultured in fresh RPMI-complete medium supplemented with IL-2 and the supernatant sampled for reverse transcriptase activity at 0, 3, 6, 10 and 14 days post inoculation, as described [87].

## Biochemical analysis of virion envelope content from primary macaque PBMCs

Rhesus macaque PBMCs were infected as for the replication assays above. Six days post-inoculation, cell-free supernatants were removed, and virions pelleted through 20% sucrose by ultracentrifugation for 120 mins. Viral pellets were resuspended in 1X TNE buffer and quantified by p27 ELISA. The samples were reduced by combining NuPAGE 10X Reducing Agent (Life

Technologies) and NuPAGE 4X LDS Sample Buffer (Life Technologies) and incubating at 95˚C for 10 mins. Equal amounts of p27 were loaded for PAGE and proteins transferred to a PVDF membrane and blocked with 5% NFM for 1 hr at RT. Membranes were cut in half and blotted separately with murine anti-Gag (3A8) or anti-gp120 (DA6) antibodies. Blots were incubated with goat anti-mouse horse radish peroxidase (HRP)-conjugated secondary antibody and the bands developed with Luminata Forte Western HRP Substrate (Merck Millipore).

## Stable cell lines

HeLa cells were transfected with plasmids encoding CD4-SIV Env chimeras [22] using TransIT-HeLaMONSTER (Mirus) and stable transfectants were selected with 400 μg/ml G418 Sulphate (Calbiochem). The stable transfectants were enriched for the expression of CD4 by FACS. Briefly, transfected HeLa cells were detached with PBS-EDTA (PBS, 5 mM EDTA, pH 7.4), washed, resuspended in ice-cold RPMI-1640 containing 1% (v/v) FBS and incubated with 5 μg/ml anti-CD4 (Q4120) for 1 hr. Subsequently, the cells were washed 3 times with RPMI/ 1% FCS, to remove unbound antibody, and then incubated with 2 μg/ml Alexa Fluor-488 conjugated secondary antibody for 1 hr at 4˚C. Finally, cells were washed once with RPMI/1% FCS, twice with PBS/1% FCS and sorted by FACS (FACSAria, Becton Dickinson). MDCKII cells were transfected by electroporation with plasmids encoding CD4-SIV Env chimeras using an Amaxa Cell Line Nucleofector™ Kit L and Amaxa Nucleofector II with settings L-05 (Lonza). Stable transfectants were selected with 400 μg/ml G418 Sulphate (Calbiochem). Alternatively, stable MDCKII cells were generated by transfection with HIV-based pseudoviruses encoding SIVmac239 Env constructs or CD4-SIV Env chimeras as controls. Briefly, VLPs were generated in HEK 293T cells by co-transfection of pCMV-d8.91, encoding HIV-1 *gag-pol* (enabling the formation of viral cores) together with the coding sequence for SIVmac239 Env cloned into the dual promoter self-inactivating vector pHRSIN-CSGWdINotI_pUb_Em, [88,89], and VSV-G (pMD2.G), using Fugene 6 (Promega).

## siRNA knockdowns

HeLa cells ($1.67 \times 10^6$ per 10 cm dish) expressing CD4-SIV Env chimeras were seeded and transfected the same day with siRNA oligonucleotides (145 nM) using Oligofectamine (Invitrogen). Oligonucleotides targeting the μ2 subunit of adaptor protein complex 2 (AP2) were used as duplexes with 3′-dTdT overhangs as previously described [90] (RNA sequences: sense, 5' GAUCAAGCGCAUGGCAGGCAU; antisense, 5' AUGCCUGCCAUGCGCUUGAUC [Dharmacon]). Cells were passaged and used for endocytosis assays and western blotting 24 hr later. To assess the knockdown efficiency, cells were lysed as described above, cleared of insoluble material and 30 μg cell protein separated by Laemmli SDS-PAGE. Gels were electroeluted onto Nitrocellulose (BioTrace NT, Pall) 1.2 mA/cm² for 1 hr at RT under semi-dry blotting conditions (25 mM Tris, 192 mM glycine, 0.1% [w/v] SDS, 20% [v/v] MeOH). Proteins were detected and imaged as described above.

## Endocytosis assay

Quantitative endocytosis assays were performed using HeLa cells expressing CD4-SIV Env chimeras. Cells ($42 \times 10^3$ cells/well in 4 or 24 well plates) were seeded 24 hr prior to use. For analysis, the cells were rapidly cooled with ice-cold media (RPMI-1640, 20 mM HEPES, 10 mM Bicarbonate and 0.2% [w/v] BSA, pH 7.0) and incubated with an anti-CD4 antibody (5 μg/ml Q4120) for 1 hr at 4˚C. Unbound antibody was washed away and endocytosis was initiated by rapidly warming with 37˚C media and subsequently stopped by rapidly cooling with

4˚C media at the indicated times. The anti-CD4 remaining at the cell surface was detected with HRP-conjugated anti-mouse IgG. Subsequently, cells were washed once with ice-cold media, twice with PBS, lysed (150 mM NaCl, 1% (v/v) Triton, 10 mM Hepes pH 7.0, and complete protease inhibitors [Roche]), and cleared of insoluble material. HRP in the cell lysate was detected by the addition of 50 μM Amplex Red (Invitrogen, in 150 mM NaCl, 1% [v/v] Triton, 200 μM $H_2O_2$, 10 mM HEPES pH 7.0) and measuring the rate of production of resorufin (Δfluorescence/min where the reaction obeys first order kinetics) with an EnVision Multilabel Reader (Perkin Elmer).

## Microscopy

HeLa cells were seeded on 13 mm, #1.5, glass coverslips 24 hr prior to use. Cells were incubated with 10 μg/ml anti-CD4 (Q4120) in media (DMEM, 1% FBS) for 3 hr at 37˚C or left untreated. Cells were washed (PBS pH 7.4 containing 0.9 mM $CaCl_2$ and 0.49 mM $MgCl_2$ [PBS++]), fixed (PBS++ containing 3% [w/v] formaldehyde) for 20 min, quenched (50 mM $NH_4Cl$ in PBS pH 7.4) for 15 min, permeabilized (0.05% [w/v] saponin, 1% [v/v] FCS in PBS pH 7.4) and immunolabeled with 10 μg/ ml anti-CD4 (Q4120) for 1.5 hr at RT followed by detection with secondary antibodies conjugated to Alexa Fluor dyes (Invitrogen). Finally, samples were washed with water and mounted on coverslips with Mowiol 4–88 (Calbiochem). Images were acquired using Nyquist criterion with a Leica TCS SPE confocal system with a 63x ACS APO/ NA 1.3 oil immersion lens and galvanometer driven stage insert and deconvolved using Huygens software.

MDCKII cells ($2.7 \times 10^5/cm^2$) were seeded on polyester Transwell 0.4 μm clear filters (Corning) and cultured for 6 days to establish polarized monolayers (trans-epithelial resistance of 80–100 Ω/cm2). Cells expressing CD4-SIV Env chimeras were washed twice (PBS++), fixed, and quenched as described above, and immunolabeled with 10 μg/ ml anti-CD4 (Q4120) for 1.5 hr at RT. Cells expressing SIVmac239 envelope protein were rapidly cooled with ice-cold media (DMEM, 1% [v/v] FBS), incubated with 480 nM human sCD4 (either D1-4 ARP6000 or D1-2 #7356) for 15 min prior to the addition of 10 μg/ml of anti-Env monoclonal (7D3). After 1 hr, the cells were washed, fixed (PBS++ containing 3% [w/v] formaldehyde) for 30 min 4˚C and quenched (50 mM $NH_4Cl$ in PBS pH 7.4) for 15 min at RT. All samples were permeabilized (0.05% [w/v] saponin, 1% [v/v] FCS in PBS++) and primary antibodies were detected with secondary antibodies conjugated to Alexa Fluor dyes (Invitrogen). Images were acquired as above. To identify the fluorescent signal from the lateral and apical membranes, cells were co-stained or E-cadherin (lateral) and EZ-Link Sulfo-NHS-S-S-Biotin (Pierce, as described above) added apically and detected with streptavidin Cy5 (Jackson). In addition, apical and basolateral membranes were also defined for CD4-SIV Env ΔGY where, after incubation with anti-CD4 antibody as described above, the membranes were immune-stained using anti-mouse conjugated to a different color Alexa Fluor dye (488 added to the basolateral side and 546 added to the apical side). Identical boxes were drawn around the basolateral and apical membrane domains using ImageJ and the ratio of basolateral to apical labelling density calculated. For display purposes images were deconvolved using Huygens software.

## Animals, viral inoculations, and sample collection

Ten pigtail macaques (PTM) were used in this study and were inoculated intravenously (i.v.) with 100 50% tissue culture infective dose (TCID$_{50}$) of SIVmac239ΔGY+R722G (n = 3) or SIVmac239ΔGY+R722G+ΔQTH (n = 3), or SIVmac239ΔGY+R722G+IRL (n = 4). All animals were MHC genotyped and haplotypes are shown in S5 Table. Before any procedure, the animals were anesthetized by intramuscular injection of ketamine hydrochloride (10 mg/kg).

Viruses were produced in HEK 293T cells transfected with plasmids containing full-length proviral DNA. Viruses were quantified by determining ($TCID_{50}$) on rhesus macaque PBMCs. Groups of pigtail macaques housed at TNPRC, infected with either SIVmac239ΔGY or SIV-mac239 and described in a previous study [18], were used for comparison. Prior to use, all animals tested negative for antibodies to SIV, simian T cell leukemia virus (STLV), and type D retrovirus and by PCR for type D retrovirus. Multiple blood samples and small intestinal biopsy samples (endoscopic duodenal pinch biopsy samples or jejunal resection biopsy samples) were collected under anesthesia (ketamine hydrochloride or isoflurane) at various times from each animal. Animals were euthanized if they exhibited a loss of more than 25% of maximum body weight, anorexia for more than 4 days, or major organ failure or medical conditions unresponsive to treatment (e.g., severe pneumonia or diarrhea) at the discretion of veterinarians.

## Quantitation of viral load in plasma

Plasma viral loads were determined at various times using a reverse transcription-PCR (RT-PCR) assay with a limit of detection of between 15 and 21 SIV RNA copies/ml [91].

## Lymphocyte isolation from intestinal tissues CD4 T cells in gut LPL

Intestinal cells were collected by endoscopic pinch biopsies of the small intestine from animals at various times. Intestinal biopsy procedures and isolation of cells from intestinal tissues were described previously [92]. Intestinal cells were isolated using EDTA-collagenase digestion and Percoll density gradient centrifugation.

## SGS analysis

Single genome amplification and Sanger sequencing (SGS) was performed on plasma samples from infected PTM at various time points after infection. The entire env gene was sequenced using a limiting-dilution PCR to ensure that only one amplifiable molecule was present in each reaction mixture, as described [32,93]. Sequence alignments were generated with Geneious and presented as highlighter plots (www.hiv.lanl.gov) or alignment tables generated in DIVEIN [94]. APOBEC signature mutations were identified with Hyper-Mut (www.hiv.lanl.gov).

## Statistical analysis

Statistical analyses were performed with GraphPad Prism v6.0g (GraphPad Software, Inc., La Jolla, CA). Pairwise comparisons were conducted using non-parametric tests (Kruskal-Wallis or Friedman tests), as samples sizes were insufficient to assess normality. Dunnett's or Holm-Sidak tests were used to adjust p-values for multiple comparisons. Where applicable, results are expressed as mean ± standard error of mean.

## Supporting information

**S1 Fig. Effects of the acquired ΔQTH mutations on *env*, *rev* and *tat* open reading frames.** Top Panel shows amino acid (a.a.) and nucleotide (nt) sequences for SIVmac239 and ΔGY Env, Tat and Rev proteins and mRNAs aligned by sequences in the Env CT. Known splice acceptor sites A7 and A8 are indicated for Rev and Tat with partial a.a. and nt sequences shown for the 1st exons of these proteins in blue and green, respectively. Bottom Panels show the deletion of QTH in Env (ΔQTH) that occurred in two rhesus macaques (RM) infected with SIVmac239ΔGY that progressed to AIDS [32]. In RM DT18 ΔQTH resulted from loss of

nt 8803–8811 and generated a new YFQI sequence in Env; in RM DD84 ΔQTH resulted from loss of nt 8804–8812 and generated a new YFQL sequence. Both ΔQTH mutations occur in regions of splice acceptor sites utilized for second exons of *rev* and *tat*. The effects of these mutations on Rev and Tat mRNA were subsequently determined *in vitro* and *in vivo* and are shown in S8 Fig. Both ΔQTH mutations arose in association with the R722G mutation shown flanking the ΔGY mutation site.
(EPS)

**S2 Fig. The ΔGY mutation and mutations acquired *in vivo* modulate cellular distribution and trafficking. (A)** Steady state cellular distribution of CD4-SIV Env CD chimeras in HeLa cells (described in Figs 3, 5 and 8). **(B)** Cellular distribution of CD4-SIV Env CD chimeras in HeLa cells after incubation with anti-CD4 (Q4120) at 37˚C for 3 hrs prior to fixation. Confocal Z stacks were deconvolved and displayed as maximum projections. Scale bar = 10 μm. **(C)** Single confocal sections through the top, middle and bottom (surface attached to the coverslip) of cells expressing CD4-SIV Env constructs used in Fig 3 to show cell surface versus intracellular distributions. The cells were labelled as described for **(B)**.
(EPS)

**S3 Fig. The ΔQTH mutations are AP2 dependent endocytic motifs.** HeLa cells expressing CD4-SIV Env CD chimeras (Fig 3) were transfected with siRNA targeting the μ2 subunit of adaptor-related protein complex 2 (AP2). **(A)** A representative western blot of cell lysates incubated with antibodies to AP2 subunits (α-adaptin or μ2) or with a loading control (anti-VDAC). **(B)** Quantitation of western blots. AP2 subunit depletion was calculated by comparison to mock conditions for all cell lines. Efficient μ2 depletion was achieved (69 ± 12%); destabilization of the AP2 complex was demonstrated by a 53 ± 13% reduction of the α-adaptin subunit. **(C)** Endocytic rates of CD4-SIV Env truncated CD constructs +/- μ2 siRNA transfection. Results are expressed as the rate of CD4 endocytosis over the first 5 mins after warming up. Graphs show the mean from n≥ 3 independent experiments.
(EPS)

**S4 Fig. *In vitro* replication of SIVmac239 containing the ΔGY mutation with or without mutations that were acquired *in vivo*.** PBMCs from rhesus or pigtail macaques were activated with ConA and IL-2 and infected with SIVmac239, ΔGY, or ΔGY containing the indicated mutations: R722G, ΔQTH creating YFQI (ΔQTH YFQI), ΔQTH creating YFQL (ΔQTH YFQL), or R722G in combination with either ΔQTH (YFQI) or ΔQTH (YFQL). Reverse transcriptase (RT) activity in culture supernatants was measured at the indicated time points. Selected data from 3 separate experiments are shown. SIVmac239ΔGY viruses containing only a ΔQTH mutation replicated poorly but were rescued by addition of R722G.
(EPS)

**S5 Fig.** Gut CD4 T cells in pigtail macaques inoculated with SIVmac239ΔGY virus containing R722G with or without a ΔQTH or IRL mutation.
(EPS)

**S6 Fig.** Platelet counts in pigtail macaques inoculated with SIVmac239ΔGY virus containing R722G with or without a ΔQTH or IRL mutation set.
(EPS)

**S7 Fig. Single genome amplification sequence analyses of plasma viral RNA from pigtail macaques inoculated with SIVmac239ΔGY virus containing R722G with or without a ΔQTH mutation.** SGA sequencing of plasma virus from the indicated time points is shown for animals inoculated with SIVmac239ΔGY containing both R722G and ΔQTH mutations (Panels **A**, **B**

and **C**), or ΔGY containing R722G (Panels **D**, **E** and **F**). Amino acid sequences (a.a. 701–879) are shown for the Env distal membrane spanning domain and the entire cytoplasmic tail. Amplicons are shown relative to parental SIVmac239 with a.a. identity indicated by a period and deletion mutations indicated by a dash. Plasma viral loads over time for each animal are shown. (PDF)

**S8 Fig. Splicing sites for Rev and Tat mRNAs used *in vivo* during infection with SIV-mac239ΔGY variants.** mRNA splicing sites for Rev and Tat mRNAs were determined on PBMCs at Day 14 after pigtail macaques were infected with SIVmac239ΔGY containing R722G with or without the ΔQTH mutation that generated the YFQL sequence in Env (**Pink Box**). SGA was used to amplify regions of Rev and Tat mRNAs flanking the predicted splice sites. **(A) Top Panel** shows a.a. and nt sequences for SIVmac239 Env and Tat proteins with *tat* splice acceptor sites A7 and A8 indicated, along with corresponding partial a.a. and nt sequences. Sequences from *tat* exon 1 are shown in **Green**. Amino acid sequences for *tat* splic-ing variants are shown (**Yellow Box**). **Middle Panel** shows results for the SIVmac239ΔGY +R722G virus; **Lower Panel** shows results for the SIVmac239ΔGY+R722G+ΔQTH (YFQL) virus. Animal identifiers (see Figs 7 and 9) and the number of amplicons exhibiting the indi-cated splicing pattern relative to the total number of amplicons are shown, as are the corre-sponding Tat a.a. sequences flanking the splicing sites. Novel variants that were generated are shown and indicated by an asterisk (*) **(B)** A similar representation is shown for Rev mRNA splicing patterns for these viruses. Amino acids from Rev exon 1 are shown in **Blue**. (EPS)

**S9 Fig. Effects of the IRL Env mutations on *rev* and *tat* open reading frames. Top Panel** shows a.a. and nt sequences for SIVmac239 and ΔGY Env, Tat, and Rev as shown in S1 Fig, with known splice acceptor sites indicated and partial a.a. and nt sequences for the 1$^{st}$ exons Rev (**Blue**) and Tat (**Green**). **Bottom Panel** shows point mutations acquired in animal KV73 that was inoculated with SIVmac239ΔGY +R722G (**Magenta**) and that progressed to AIDS (Fig 6). Nt mutations are indicated in **orange** as are resulting a.a. changes in Env, Rev and Tat. **Red Arrows** show "IRL" a.a. changes T735I, Q739R, and P744L in Env. A G-to-A nt mutation is also shown that was silent in Env but produced Gly-to-Ser and Arg-to-Glu mutations in Rev and Tat, respectively. The C-to-T nt change that produced P744L in Env also generated a stop codon (*) in the Tat second exon. The presence of this stop codon was confirmed on mRNAs from macaque PBMCs infected with an SIV that contained the mutations shown above. The Env R751G fitness mutation is also shown along with its predicted mutation in Rev [41]. (EPS)

**S10 Fig. Plasma viral loads for PTM infected with SIVmac239ΔGY+R722G+IRL.** PTM were infected with SIVmac239ΔGY+R722G+IRL. Plasma viral loads are shown. Arrows denote time points at which plasma was obtained for SGS. Animal identifiers are indicated. (EPS)

**S11 Fig. SGS of plasma viral RNA from pigtail macaques inoculated with SIVmac239ΔGY +R722G+IRL.** SGS of plasma virus from the indicated time points is shown for 4 animals inoc-ulated with SIVmac239ΔGY containing R722G and 3 point mutations (T735I, Q739R, and P744L) shown to confer a new basolateral sorting signal (Fig 8). Amino acid sequences are shown for the Env distal membrane spanning domain and the entire cytoplasmic tail. Ampli-cons are shown relative to parental SIVmac239 with a.a. identity indicated by a period and deletions indicated by a dash. Plasma viral loads over time for each animal are shown. (PDF)

**S1 Table. Viral evolution at the Env C-terminus in pigtail macaques inoculated with SIV-mac239ΔGY containing the R722G and ΔQTH mutations.** Summary of single genome amplification of plasma virus is shown for 3 pigtail macaques inoculated i.v. with the SIV-mac239ΔGY+R722G+ΔQTH virus. Sequences for SIVmac239, ΔGY and parental ΔGY +R722G +ΔQTH virus are shown at the top. Results are shown for mutations within a.a. 830 to the C-terminus at position 880. The numbers of amplicons bearing the indicated mutations and the % of amplicons are shown. Dashes indicates identity with ΔGY Env. See S7 Fig for a complete listing of sequences of individual amplicons.
(EPS)

**S2 Table. Viral evolution at the Env C-terminus in pigtail macaques inoculated with SIV-mac239ΔGY containing the R722G mutation.** Summary of single genome amplification analysis of plasma virus performed at the indicated time points for 3 pigtail macaques inoculated i.v. with SIVmac239ΔGY+R722G virus. Results are shown for mutations within a.a. 830 to the C-terminus at position 880 as in S1 Table. Sequences for parental SIVmac239, ΔGY and ΔGY +R722G Envs are shown at the top. Complete listings of a.a. sequences in the cytoplasmic tail for individual amplicons are shown in S7 Fig.
(EPS)

**S3 Table. Viral evolution in PTM inoculated with SIVmac239ΔGY+R722G+IRL.** Summary of SGS of plasma virus performed at the indicated time points for 4 PTM inoculated with SIV-mac239ΔGY+R722G+IRL. Sequences for SIVmac239, SIVmac ΔGY and SIVmac239ΔGY +R722G+IRL are shown at the top. Complete listings of CD a.a. sequences for individual amplicons are shown in S11 Fig.
(EPS)

**S4 Table. Viral evolution at the Env C-terminus in PTM inoculated with SIVmac239ΔGY +R722G+IRL.** Summary of SGS analysis of plasma virus performed at the indicated time points for 4 PTMs inoculated i.v. with the SIVmac239ΔGY+R722G virus containing the IRL set that arose in animal KV53 and conferred a new basolateral sorting signal (see Fig 8). Results are shown for a.a. 800 to the C-terminus at position 880 as in Figs 7 and 9. Sequences for SIV-mac239, ΔGY and parental ΔGY+R722G+IRL Envs are shown at the top. Complete listings of a.a. sequences in the cytoplasmic domains for individual amplicons are shown in S11 Fig.
(EPS)

**S5 Table. MHC-I haplotypes of pigtails used in this study.** MHC for SIVmac239 and SIV-mac239ΔGY infected animals were shown in a previous manuscript [18] (n/d, not determined).
(EPS)

## Acknowledgments

We thank Drs Catherine Hogan (Cardiff University) and Karl Matlin (University of Chicago) for advice and protocols for working with MDCK cells and Graham Warren (UCL) for critical comments on the manuscript.

## Author Contributions

**Conceptualization:** Scott P. Lawrence, Samra E. Elser, Pyone P. Aye, Celia C. LaBranche, Andrew A. Lackner, Nicholas J. Maness, Mark Marsh, James A. Hoxie.

**Data curation:** Scott P. Lawrence, Samra E. Elser, Workineh Torben, Robert V. Blair, Faith Schiro, Lara A. Doyle-Meyers, Beth S. Haggarty, Andrea P. O. Jordan, Josephine Romano, George J. Leslie, Xavier Alvarez, Christine M. Fennessey, Yuan Li, Jeffrey D. Lifson, Celia C. LaBranche, Brandon F. Keele, Nicholas J. Maness, Mark Marsh, James A. Hoxie.

**Formal analysis:** Scott P. Lawrence, Samra E. Elser, Workineh Torben, Robert V. Blair, Bapi Pahar, Dawn Szeltner, Lara A. Doyle-Meyers, Xavier Alvarez, David H. O'Connor, Roger W. Wiseman, Christine M. Fennessey, Yuan Li, Jeffrey D. Lifson, Brandon F. Keele, Nicholas J. Maness.

**Funding acquisition:** Mark Marsh, James A. Hoxie.

**Investigation:** Scott P. Lawrence, Samra E. Elser, Workineh Torben, Robert V. Blair, Faith Schiro, Dawn Szeltner, Lara A. Doyle-Meyers, Beth S. Haggarty, Andrea P. O. Jordan, Josephine Romano, George J. Leslie, Xavier Alvarez, Christine M. Fennessey, Yuan Li, Jeffrey D. Lifson, Brandon F. Keele, Nicholas J. Maness, James A. Hoxie.

**Methodology:** Scott P. Lawrence, Samra E. Elser, Workineh Torben, Robert V. Blair, Lara A. Doyle-Meyers, Beth S. Haggarty, Andrea P. O. Jordan, Josephine Romano, George J. Leslie, Xavier Alvarez, David H. O'Connor, Roger W. Wiseman, Christine M. Fennessey, Yuan Li, Michael Piatak, Jr, Jeffrey D. Lifson, Brandon F. Keele, Nicholas J. Maness, Mark Marsh, James A. Hoxie.

**Project administration:** Scott P. Lawrence, Pyone P. Aye, Lara A. Doyle-Meyers, Jeffrey D. Lifson, Andrew A. Lackner, Brandon F. Keele, Nicholas J. Maness, Mark Marsh, James A. Hoxie.

**Resources:** David H. O'Connor, Roger W. Wiseman.

**Supervision:** Bapi Pahar, Pyone P. Aye, Faith Schiro, Jeffrey D. Lifson, Celia C. LaBranche, Mark Marsh, James A. Hoxie.

**Validation:** Scott P. Lawrence, Jeffrey D. Lifson, Brandon F. Keele, Nicholas J. Maness, Mark Marsh.

**Visualization:** Scott P. Lawrence, Jeffrey D. Lifson, Brandon F. Keele, Mark Marsh.

**Writing – original draft:** Scott P. Lawrence, Nicholas J. Maness, Mark Marsh, James A. Hoxie.

**Writing – review & editing:** Scott P. Lawrence, David H. O'Connor, Jeffrey D. Lifson, Brandon F. Keele, Nicholas J. Maness, Mark Marsh, James A. Hoxie.

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
