## [Decision Letter · Decision Letter 0]

17 Jan 2022

Dear Dr. Marsh,

Thank you very much for submitting your manuscript "A cellular trafficking signal in the SIV envelope protein cytoplasmic domain is strongly selected for in pathogenic infection" for consideration at PLOS Pathogens. As with all papers reviewed by the journal, your manuscript was reviewed by members of the editorial board and by several independent reviewers. The reviewers appreciated the attention to an important topic. Based on the reviews, we are likely to accept this manuscript for publication, providing that you modify the manuscript according to the review recommendations.

Sincerely,

Daniel C. Douek

Associate Editor

PLOS Pathogens

Richard Koup

Section Editor

PLOS Pathogens

Kasturi Haldar

Editor-in-Chief

PLOS Pathogens

orcid.org/0000-0001-5065-158X

Michael Malim

Editor-in-Chief

PLOS Pathogens

orcid.org/0000-0002-7699-2064

Reviewer Comments (if any, and for reference):

Reviewer's Responses to Questions

**Part I - Summary**

Reviewer #1: Previous studies by this group have demonstrated that deletion of two residues at Env 720 and 721 (�GY) within a highly conserved tyrosine-dependent trafficking motif in the cytoplasmic tail of SIVmac239 resulted in dramatically reduced pathogenesis compared to wildtype infection of pigtail macaques. This group has also reported that SIVmac239 can acquire proximal changes in vivo, including an adjacent substitution (R722G) and a nearby three residue deletion (�QTH), that can contribute to disease progression, in pigtail and rhesus macaques. Here the authors performed a series of detailed in vitro and in vivo experiments to parse out the contributions of these acquired changes to endocytosis and polarized sorting of Env, as well as their roles in pathogenesis. The results show that both are maintained in �GY viruses in vivo, with �QTH being key to restoring endocytosis and basolateral sorting functions. Interestingly, �QTH, which occurs near splice junctions and thus impacts the overlapping Tat and Rev reading frames, generates a de novo tyrosine dependent motif. The R722G substitution was found to be critical for replication fitness in the context of �QTH and may contribute to increasing Env virion content. In addition, a novel set of three substitutions (IRL) was observed in the context of �GY + R722G infection in animals that exhibited signs of disease progression. These changes were also completely conserved and found to reconstitute sorting but not endocytosis function. Remarkably, the L substitution creates a premature stop codon in Tat. The results overall suggest that endocytosis may have redundant signals in the tail, while the GYxx� motif destroyed by �GY may be the principal regulator of basolateral sorting, a function under strong selective pressure. This detailed and comprehensive study demonstrates the strong selective pressure for proper Env trafficking in vivo, shedding new light onto underappreciated mechanisms used by HIV/SIV for persistence, immune evasion, cell to cell transmission, and ultimately disease progression.

The study is important for several reasons and should appeal to a broad audience of readers. One is that the unusually long cytoplasmic tail of HIV/SIV is poorly understood, and it contains multiple and potentially redundant motifs that are associated with such functions as endocytosis, trafficking, and polarized sorting. Not many studies have been carried out to carefully probe these functions in vivo. Another is that polarized sorting may be important for cell-to-cell transmission by affecting viral tropism and spreading in sites such as the gut. Third is that the loss of Env trafficking functions appears to make the virus more vulnerable to immune control, highlighting potential avenues for therapeutic intervention and cure strategies. Furthermore, the acquisition of compensatory mutations in the setting of �GY is a fascinating display of selective pressure, maintenance of multiple overlapping open reading frames, and convergent viral evolution pathways by this virus.

Reviewer #2: Lawrence et al. studied the role of a highly conserved GYxxØ trafficking signal in the cytoplasmic domain of SIVmac239 envelope glycoprotein (Env) in viral infection and pathogenesis. The team previously reported that SIVmac239 containing a deletion in the GYxxØ signal (SIVmac239ΔGY) can replicate in rhesus (RM) and pigtailed macaques (PTM) to high levels, but is rapidly controlled (especially in the latter). However, revertants of SIVmac239ΔGY in RM and PTM that progressed to AIDS were found to be associated with mutations in R722 flanking the ΔGY deletion and a nine-nucleotide deletion encoding amino acids 734-736 (ΔQTH) that overlaps the rev and tat open reading frames. In this manuscript, the authors dissected the role of these mutations to restore the Env trafficking functions ablated by the ΔGY deletion. The authors also identified a novel genotype (“IRL”) in PTM infected with SIVmac239ΔGY+R722G and demonstrated its ability to restore the polarized sorting function of GYxxØ, but not endocytosis. This genotype was highly conserved when introduced into naïve PTMs. The authors suggested that the strong selection for Env endocytosis and particularly for polarized sorting during pathogenic SIV infection highlights the critical roles played by these cellular trafficking functions in HIV/SIV pathogenesis.

This is a well-conceived study addressing a central issue of the role of Env trafficking in HIV/SIV infectivity and pathogenesis. The overall approach was well-reasoned and the experiments well designed. The finding that polarized sorting of Env may play an important role in HIV/SIV pathogenesis is novel and has important implications. The main conclusions were supported by the findings and the manuscript clearly written.

Reviewer #3: The authors here investigate the significance of compensatory changes occurring in the cytoplasmic tail of SIV Env in pigtail or rhesus macaques infected with the deltaGY mutant of SIVmac239. deltaGY has been of interest because pigtail macaques control infection by this virus, and because this mutant disrupts a known Yxx� trafficking signal involved in endocytosis and basolateral sorting of Env. The authors show here that endocytosis and sorting signals are reconstituted, most evidently through a deletion that creates a new Yxx� sequence, but also through a previously unknown combination of changes (“IRL set”) that restores sorting but not endocytosis. The data here are extensive and convincing regarding the restoration of endocytosis and sorting signals. The association of restoration of sorting with restored pathogenicity is highly suggestive that sorting through signals in the cytoplasmic tail is an important determinant of in vivo replication and escape from immune control, although the exact mechanism is not defined. Overall, this is an important contribution that should stimulate additional work to define how a sorting signal determines pathogenicity. The combination of macaque pathogenesis work and bask cellular trafficking studies here is remarkable and exciting.

**Part II – Major Issues: Key Experiments Required for Acceptance**

Reviewer #1: The methods are described in detail and performed with rigor.

No substantial weaknesses were noted in this study.

Reviewer #2: None

Reviewer #3: None

**Part III – Minor Issues: Editorial and Data Presentation Modifications**

Reviewer #1: Minor points are the use of historical controls, and the question of whether SIVmac239 (a widely accepted model virus for HIV pathogenesis) is completely reflective of other SIV/HIV viruses. However, these are minor points that do not detract from this overall laudable study.

Reviewer #2: (No Response)

Reviewer #3: 1. Figure 2 shows a very reasonable measurement of cell surface and virion Env for the �GY and revertant viruses by surface biotinylation, pull down, and immunoblotting. Flow cytometry data establishing levels of cell surface Env would add to this as a direct quantification of cell surface Env levels for deltaGY and the revertant viruses.

2. Figure 3B demonstrates subcellular localization by immunofluorescence microscopy. While some differences in subcellular localization can be appreciated as shown, the use of maximum projections does not allow the conclusion that the diffuse signal seen with deltaGY, deltaGY +R722G, deltaGY +S727P is on the cell surface. A supplemental figure with a confocal section at the coverslip level vs. mid-cell level would better establish this PM distribution.

PLOS authors have the option to publish the peer review history of their article (what does this mean?). If published, this will include your full peer review and any attached files.

Reviewer #1: No

Reviewer #2: No

Reviewer #3: **Yes: **Paul Spearman

Figure Files:

Data Requirements:

Reproducibility:

References:

---

## [Decision Letter · Decision Letter 1]

7 Apr 2022

Dear Dr. Marsh,

We are pleased to inform you that your manuscript 'A cellular trafficking signal in the SIV envelope protein cytoplasmic domain is strongly selected for in pathogenic infection' has been provisionally accepted for publication in PLOS Pathogens.

Best regards,

Daniel C. Douek

Associate Editor

PLOS Pathogens

Richard Koup

Section Editor

PLOS Pathogens

Kasturi Haldar

Editor-in-Chief

PLOS Pathogens

orcid.org/0000-0001-5065-158X

Michael Malim

Editor-in-Chief

PLOS Pathogens

orcid.org/0000-0002-7699-2064

Reviewer Comments (if any, and for reference):

Reviewer's Responses to Questions

**Part I - Summary**

Reviewer #2: The authors have adequately addressed all the concerns from the previous reviewers and have revised the manuscript accordingly. The revised manuscript is acceptable for publication.

Reviewer #3: This report was already quite strong and will be important to the field. The response is entirely fine including the added supplemental figure.

**Part II – Major Issues: Key Experiments Required for Acceptance**

Reviewer #2: None

Reviewer #3: (No Response)

**Part III – Minor Issues: Editorial and Data Presentation Modifications**

Reviewer #2: None

Reviewer #3: (No Response)

PLOS authors have the option to publish the peer review history of their article (what does this mean?). If published, this will include your full peer review and any attached files.

Reviewer #2: No

Reviewer #3: **Yes: **Paul Spearman

---

## [Editor Report · Acceptance letter]

25 May 2022

Dear Dr. Marsh,

We are delighted to inform you that your manuscript, "A cellular trafficking signal in the SIV envelope protein cytoplasmic domain is strongly selected for in pathogenic infection," has been formally accepted for publication in PLOS Pathogens.

Best regards,

Kasturi Haldar

Editor-in-Chief

PLOS Pathogens

orcid.org/0000-0001-5065-158X

Michael Malim

Editor-in-Chief

PLOS Pathogens

orcid.org/0000-0002-7699-2064